

# Investigation of the $\beta$-pinene photooxidation by $\mathrm{OH}$ in the atmosphere simulation chamber SAPHIR

Martin Kaminski[1,*], Hendrik Fuchs[1], Ismail-Hakki Acir[1,**], Birger Bohn[1], Theo Brauers[1,†], Hans-Peter Dorn[1], Rolf Häseler[1], Andreas Hofzumahaus[1], Xin Li[1,***], Anna Lutz[2], Sascha Nehr[1,****], Franz Rohrer[1], Ralf Tillmann[1], Luc Vereecken[1], Robert Wegener[1], and Andreas Wahner[1]

[1]Institute of Energy and Climate Research, IEK-8: Troposphere, Forschungszentrum Jülich GmbH, Jülich, Germany
[2]Department of Chemistry and Molecular Biology, University of Gothenburg , Gothenburg, Sweden
*now at: Bundesamt für Verbraucherschutz, Abteilung 5 - Methodenstandardisierung, Referenzlaboratorien und Antibiotikaresistenz, Berlin, Germany
**now at: Institute of Nutrition and Food Sciences, Food Chemistry, University of Bonn, Bonn, Germany
***now at: State Key Joint Laboratory of Environmental Simulation and Pollution Control, College of Environmental Sciences and Engineering, Peking University, Beijing, China
****now at: Verein Deutscher Ingenieure e.V., Kommission Reinhaltung der Luft, Düsseldorf, Germany
†Deceased

*Correspondence to:* Robert Wegener (r.wegener@fz-juelich.de)

**Abstract.**

Beside isoprene, monoterpenes are the non-methane volatile organic compounds (VOC) with the highest global emission rates. Due to their high reactivity towards OH, monoterpenes can dominate the radical chemistry of the atmosphere in forested areas. In the present study the photochemical

degradation mechanism of $\beta$-pinene was investigated in the Jülich atmosphere simulation chamber SAPHIR. The focus of this study is on the OH budget in the degradation process. Therefore the SAPHIR chamber was equipped with instrumentation to measure radicals (OH, $HO_2$, $RO_2$), the total OH reactivity, important OH precursors ($O_3$, HONO, HCHO), the parent VOC $\beta$-pinene, its main oxidation products, acetone and nopinone, and photolysis frequencies. All experiments

were carried out under low $NO_x$ conditions ($\leq 2\,\mathrm{ppb}$) and at atmospheric $\beta$-pinene concentrations ($\leq 5\,\mathrm{ppb}$) with and without addition of ozone. For the investigation of the OH budget, the OH production and destruction rates were calculated from measured quantities. Within the limits of accuracy of the instruments, the OH budget was balanced in all $\beta$-pinene oxidation experiments. However, even though the OH budget was closed, simulation results from the Master Chemical

Mechanism 3.2 showed that the OH production and destruction rates were underestimated by the model. The measured OH and $HO_2$ concentrations were underestimated by up to a factor of two whereas the total OH reactivity was slightly overestimated because of the poor reproduction of the measured nopinone by the model by up to a factor of three. A new, theory-derived first-generation





product distribution by Vereecken and Peeters was able to reproduce the measured nopinone time
series and the total OH reactivity. Nevertheless the measured OH and $HO_2$ concentrations remained
underestimated by the numerical simulations. These observations together with the fact that the
measured OH budget was closed suggest the existence of unaccounted sources of $HO_2$.

## 1   Introduction

Thousands of different volatile organic compounds (VOCs) are emitted into the atmosphere (Gold-
stein and Galbally, 2007). The emissions of biogenic volatile organic compounds BVOCs exceed
those of anthropogenic VOCs by a factor of ten (Piccot et al., 1992; Guenther et al., 1995, 2012).
On a global scale, isoprene and monoterpenes are the BVOCs with the highest emission rates with
the exception of methane. About 44 % of the global BVOC emissions can be attributed to isoprene
and about 11 % to monoterpenes (Guenther et al., 1995). Isoprene and monoterpenes are unsaturated
hydrocarbons. Hence, their main atmospheric sink is the addition of hydroxyl radicals (OH), nitrate
radicals ($NO_3$) or ozone to the double bond (Calogirou et al., 1999; Atkinson and Arey, 2003). Dur-
ing daytime the reaction of isoprene and monoterpenes with the OH radical is the major sink for
these VOC species. The subsequent addition of oxygen produces organic peroxy radicals ($RO_2$). In
the presence of nitrogen oxides ($NO_x$), $RO_2$ is indirectly converted to hydroperoxy radicals ($HO_2$)
through reaction with NO. $HO_2$ reacts further with NO in a second reaction step to OH, recyling
the consumed OH from the initial reaction step and producing further $NO_2$. As a side effect, ozone
is produced by $NO_2$ photolysis. The oxidation of VOCs in the presence of $NO_x$ is the main photo-
chemical source of ozone in the troposphere (Seinfeld and Pandis, 2006). Moreover, the oxidation
processes of isoprene and monoterpenes mainly lead to the production of less reactive polar oxy-
genated volatile organic compounds (OVOCs) which are significantly involved in the formation of
secondary organic aerosols (SOA).

As trees have the largest contribution in global non-methane BVOC emissions with an estimated
75 % (Guenther et al., 1995), our understanding of VOC oxidation processes in forested regions is
essential for the understanding of the BVOC oxidation processes on a global scale (Guenther et al.,
1995; Wiedinmyer et al., 2004; Guenther et al., 2012). Therefore the BVOC degradation by OH rad-
icals in forested regions has been investigated in numerous field campaigns (Lelieveld et al., 2008;
Hofzumahaus et al., 2009; Kubistin et al., 2010; Whalley et al., 2011; Lu et al., 2012) over the last
decades. The results showed a lack of knowledge about photochemical oxidation processes under
low $NO_x$ conditions and high BVOC concentrations in these regions (Rohrer et al., 2014). State of
the art atmospheric chemistry models were largely underestimating the measured OH concentra-
tions. These observations could be partially explained by the identification of new NO independent
OH recycling pathways in the oxidation of isoprene through lab experiments and chamber studies
in the following years (Paulot et al., 2009; da Silva et al., 2009; Peeters and Müller, 2010; Crounse



et al., 2011, 2012; Wolfe et al., 2012; Crounse et al., 2013; Liu et al., 2013; Fuchs et al., 2013, 2014;
Peeters et al., 2014). Since the field campaigns were mainly conducted in tropical forests, where VOC emissions are dominated by isoprene, most mechanistic investigations focused on isoprene. Regardless of the integration of new NO independent recycling pathways in the isoprene degradation mechanism, models could not completely explain the measured OH concentrations.

Due to their high abundance and their structural similarity to isoprene, monoterpenes may con-
60 tribute to an underestimation of the OH concentration in the models as proposed by da Silva et al. (2009) for open chain monoterpenes like myrcene and ocimene. Moreover, during a field campaign in Borneo Whalley et al. (2011) observed that discrepancies between measured and modeled OH also occurred in the morning hours when VOC emissions were dominated by monoterpenes. Recent studies in Finland (Hens et al., 2014) and the U.S. (Kim et al., 2013) indicate that the radical
chemistry in forested areas that are dominated by monoterpene and 2-methyl-3-buten-2-ol (MBO) emissions is not well understood.

In this work we investigated the atmospheric degradation of monoterpenes in the SAPHIR atmosphere simulation chamber in Jülich. $\beta$-Pinene comprises 17 % of the estimated global monoterpene emission rate (Sindelarova et al., 2014) and was therefore chosen as a representative species for
our investigations. To our knowledge it is the first chamber study investigating the $\beta$-pinene, or any monoterpene degradation in general, under natural concentration conditions (VOC less than 5 ppb). In comparison to other chamber studies which focused on the determination of products and SOA yields (Lee et al., 2006; Saathoff et al., 2009; Eddingsaas et al., 2012a, b; Zhao et al., 2015) our main goal was to investigate the radical budget of the monoterpene degradation. For that purpose all
critical radical species (OH, HO$_2$, RO$_2$) were quantified.

## 2 Methods

### 2.1 SAPHIR atmosphere simulation chamber

The atmosphere simulation chamber SAPHIR (Simulation of Atmospheric PHotochemistry In a large Reaction Chamber) located in the Forschungszentrum Jülich (Germany) is a tool to investigate
complex atmospheric mechanisms under nearly natural conditions. The chamber has a cylindric shape (18 m length, 5 m diameter, 270 m$^3$ volume) and consists of a double walled FEP Teflon foil attached to a steel frame. The Teflon foil guarantees a maximum of inertness of the chamber surface and leads to a minimization of wall effects. In SAPHIR natural sunlight is used as light source for photochemical reactions. About 85 % of the UV-A, UV-B and visible light is transmitted by the FEP
foil. A shutter system allows to switch between illuminated and dark chamber conditions within 60s. To investigate photochemical degradation processes in the ppb and sub-ppb range SAPHIR is operated with ultra pure synthetic air (Linde, N$_2$ 99.9999 %, O$_2$ 99.9999 %). A slight overpressure of about 30 Pa in the inner chamber prevents diffusion of outside air into SAPHIR. Due to small





leakages and consumption of air by instruments a replenishment flow has to be introduced into
the chamber to keep up the pressure difference to the outside. During experimental operation this
flow is in a range of $9\text{-}12\,\mathrm{m^3h^{-1}}$, leading to a dilution of trace gases at a rate of approximately
$3\text{-}4\,\%\ \mathrm{h^{-1}}$. An installed ventilator guarantees well mixed conditions during the experiments. For
more detailed information about the chamber and its properties the reader is referred to previous
publications (Poppe et al., 2007; Schlosser et al., 2007, 2009; Wegener et al., 2007; Dorn et al.,
2013).

### 2.2 Instrumentation

To investigate the radical budget during the $\beta$-pinene photooxidation, SAPHIR was equipped with
a comprehensive set of analytical instruments. OH, $HO_2$ and $RO_2$ concentrations were measured
simultaneously by a laser-induced fluorescence system, using three independent measurement cells.
$RO_2$ and $HO_2$ are both detected by conversion with NO to OH and differentiated from each other
by their different conversion efficiency with NO. Because of the short residence time in the $HO_x$
cell between NO addition and OH detection, $RO_2$ species (e.g. acyl peroxy radicals), which require
conversion to alkoxy radicals with subsequent slow transformation with $O_2$ to $HO_2$ radicals before
OH is formed, are assumed to be converted with very low efficiency. However, $RO_2$ species as
$\beta$-hydoxy alkyl peroxy radicals are converted by NO to highly reactive $\beta$-hydroxyalkoxy radicals.
Instead of reacting with $O_2$ directly $\beta$-hydroxyalkoxy radicals nearly exclusively decompose and
then react rapidly with $O_2$ forming thereby $HO_2$ much faster than other alkyl alkoxy radicals. The
fact that for $\beta$-hydroxy alkyl peroxy radicals the overall conversion to OH is thereby much faster
than for other alkyl peroxy radicals leads to an interference of this $RO_2$ subclass in the $HO_2$ channel
of the LIF instrument (Fuchs et al., 2011). The interference was carefully characterized for $RO_2$
species formed by the initial reaction of $\beta$-pinene with OH by adding $\beta$-pinene to the gas of the OH/
$HO_2$ calibration cell. OH produced by water vapour photolysis reacts with $\beta$-pinene and produces
first-generation $RO_2$ radicals. About 25 % of these $RO_2$ species are detected as an additional signal
in the $HO_2$ channel of the instrument. Since the $RO_2$ concentration is calculated from the measured
$HO_2$ concentration, the interference also effects the $RO_2$ data.

On 27th of August 2012 OH was measured additionally by a differential optical absorption spec-
trometer (DOAS). In general both instruments showed a good agreement over the past 10 years
(Schlosser et al., 2007, 2009; Fuchs et al., 2012). Also for the terpenoid campaign in 2012 on av-
erage no significant difference between LIF and DOAS instrument was observed. As the DOAS
instrument is the only absolute method for the quantification of OH, the DOAS OH data were used
for the following evaluation of the OH budget analysis.

The OH reactivity $k(OH)$ was measured by flash photolysis / laser induced fluorescence (FP/LIF)
technique (Lou et al., 2010). The evaluation of the pseudo-first-order decays of OH gives a direct
measure of the total rate coefficient of the OH loss.



Besides OH, HO$_2$, RO$_2$ and $k$(OH), VOC (proton transfer reaction time of flight mass spec-
       trometry PTR-TOF-MS, gas chromatography coupled with mass spectrometric and flame ioniza-
       tion detector (GC/MS/FID), HCHO (Hantzsch reaction), HONO (long path absorption photome-
       try, LOPAP), CO (reduction gas analysis, RGA), CO$_2$, CH$_4$, H$_2$O (cavity ring-down spectroscopy,
       CRDS), as well as NO, NO$_2$ and O$_3$ (chemiluminescence, CL) were determined by direct measure-
ments. Moreover experimental boundary conditions including temperature (ultrasonic anemometer),
       pressure (capacitive gauge), replenishment flow rate (mass flow controller) and photolysis frequen-
       cies (spectroradiometer) were continuously recorded.

       Table 1 provides an overview of the key instruments for this study and their specifications. For
       more detailed information on the analytical instrumentation of SAPHIR the reader is referred to
previous publications ((Bohn and Zilken, 2005; Rohrer et al., 2005; Wegener et al., 2007; Dorn
       et al., 2013)), and references therein.

### 2.3    Experimental procedure

Before every experiment day the chamber was flushed with dry ultra-pure synthetic air over night
to purge contaminants of previous experiments under their detection limit. At the beginning of the
experiment 20 ppm of CO$_2$ were injected into SAPHIR as dilution tracer. After that the relative
       humidity was increased to 75 % by adding water vapour, generated by the vaporisation of ultra-pure
       water (Milli-Q), to the purge flow. As HONO photolysis is the main source of OH in the SAPHIR
       chamber it is impossible to conduct experiments in the complete absence of NO. To lower the NO
       level in the experiment on 27th August 50 ppb of ozone, produced from a silent discharge ozonizer
(O3Onia), was injected after humidification. Shortly afterward the shutter system of SAPHIR was
       opened, exposing the chamber to sunlight.

       In the following two hours of the experiments (so-called "zero air phase") no other trace gases
       were introduced into SAPHIR. During the zero air period HONO was formed from the chamber
       walls (Rohrer et al., 2005) depending on relative humidity and UV radiation. In addition to the OH
production the photolysis of HONO leads to an increase in NO and NO$_2$ concentration. In addition
       acetaldehyde, formaldehyde and acetone were formed in the chamber with a rate of $90 - 250$ ppt/h.
       The zero air phase ended with the injection of $\beta$-pinene while the SAPHIR chamber was exposed
       to light.. The injection was performed by introducing a high concentration gas mixture of $\beta$-pinene
       (about 50 ppm) from a silcosteel canister (Restek) through a mass flow controller to the experimental
flow. The $\beta$-pinene concentration of the mixture was previously determined by oxidizing a part of the
       $\beta$-pinene mixture on a platinum catalyst and quantifying the produced CO$_2$. This absolute method
       makes it possible to calculate the VOC starting concentration of the experiment very accurately.
       During the following 6 h of the experiment, the so-called "VOC phase", $\beta$-pinene was degraded
       by OH in the illuminated chamber. In the experiment of 27th August $\beta$-pinene was injected for a
second and third time into SAPHIR approximately two and four hours after the first VOC injection,



respectively. Every experiment ended with closing the louver system of the chamber in the late evening of the experiment day. For all the chamber experiments the fan was running during the whole time ensuring homogeneous mixing of the chamber air.

Table 2 sums up the experimental conditions of the three $\beta$-pinene oxidation experiments.

## 2.4 Model calculations

The acquired time series of trace gases and radicals were compared to model simulations with the Master Chemical Mechanism (MCM) . The MCM is a state of the art 1-dimensional near explicit atmospheric box model developed by Jenkin et al. (1997) and Saunders et al. (2003). For this publication the MCM version 3.2 was used (available at http://mcm.leeds.ac.uk/MCMv3.2/). For the application on modeling chamber experiments the model was extended by some chamber specific processes. As an alternative to the $\beta$-pinene chemistry in the MCM, we also applied the kinetic model by Vereecken and Peeters (2012), which replaces the $\beta$-pinene chemistry in the MCM based on theoretical-kinetic analyses of the reaction mechanism. The current published mechanism only describes the first-generation product formation, i.e. the subsequent chemistry of the products formed in the first radical chain is not included in the model. This could affect the predictions, in so far as these primary stable products contribute significantly to the reaction chemistry. The accumulated product yield of these primary products in our model runs remains below 20 % compared to the sum of the residual concentration of $\beta$-pinene, and the concentrations of reactive primary products whose chemistry is fully described (nopinene, acetone, ...). As such, it appears that omitting the secondary chemistry of these products does not have an overly large impact on the reaction fluxes, and are is therefore unlikely to be the main reason for any discrepancies relative to the measurements.

As mentioned in section 2.1 the required replenishment flow into SAPHIR leads to an additional dilution process for every model species. The applied dilution rate is thereby calculated from the measured $CO_2$ loss in the chamber. Previous characterization experiments showed that ozone had a shorter lifetime than $CO_2$ in the chamber (dilution corrected ozone lifetime approximately 30 h). This observation was included as additional loss term in the model. The chamber sources of HONO, HCHO and acetone are well known from routine reference experiments in SAPHIR and can be parametrized by empirical equations, depending on temperature, relative humidity and solar radiation in the chamber (Rohrer et al., 2005; Karl et al., 2006; Kaminski, 2014). The source strengths were adjusted to match the time series of $NO_x$, HCHO and acetone during the zero air phases of the experiments. The parametrization of the acetaldehyde source was less satisfactory and so the model was constrained by the measured acetaldehyde concentration.

In all experiments the summed contributions of known chamber sources to the OH reactivity measured in the zero air phase $(0.1 - 0.7 \, s^{-1})$ were not sufficient to explain the measured OH reactivity $(0.7 - 1.5 \, s^{-1})$. Analogous to the procedure applied by Fuchs et al. (2012, 2014) the unexplained part of the measured OH reactivity was modeled as a co-reactant Y, with constant OH reactivity in





the model, where the concentration time rate coefficient kOH [Y] was set to reproduce the measured OH reactivity in the chamber after humidification. Analogous to CO the reaction of Y with OH is assumed to form one molecule of $HO_2$.

The parameters temperature, pressure, water vapor concentration, the calculated dilution rate and the photolysis frequencies for HONO, HCHO, $O_3$ and $NO_2$ were set as fixed boundary conditions in the model. Photolysis frequencies that were not measured were calculated for clear sky conditions by the function included in MCM 3.1 and then corrected for cloud cover and the transmission of the Teflon film by multiplying the clear sky value with the ratio of measured to modeled photolysis

frequency of $NO_2$. Constrained parameters were re-initialized on a 1 min time grid. The injections of $\beta$-pinene and ozone in the chamber were modeled as sources which were only present during the time period of injection. The source strengths were adapted to match the measured ozone concentration and the OH reactivity at the point of injection. The subsequent time series of the concentrations were determined by the kinetic models described above.

Because of described instrumental interferences it is not possible to directly compare the modeled $HO_2$ concentration $[HO_2]$ and the sum of the concentrations of the different $RO_2$ species $[RO_2]$ against the measured time series of the LIF instrument, $[HO_2^*]$ and $[RO_2^*]$, for $HO_2$ and $RO_2$, respectively.

$$[HO_2^*] = [HO_2] + \sum [RO_{2i}] \cdot \alpha_i \qquad (1)$$

$$[RO_2^*] = [RO_2] - \sum [RO_{2i}] \cdot \alpha_i \qquad (2)$$

$\alpha_i$: cross-sensitivity

$\sum [RO_{2i}]$: inferring $RO_2$ radicals of $\beta$ pinene

$\sum [RO_{2i}] \cdot \alpha_i$: $RO_2$ interference

For a direct comparison of the measured uncorrected $HO_2$ signal $[HO_2^*]$ against the model, the modeled $HO_2$ plus an estimated $RO_2$ interference is combined to yield the model parameter $HO_2^*$ (Lu et al., 2012). Depending on the experimental phase, up to 25 % of the modeled $HO_2^*$ can be attributed to the interfering $RO_2$ species $[RO_{2i}]$. Moreover, note that the MCM and the modifications by Vereecken and Peeters yield different $RO_2$ species, which results in rather different contributions

of $RO_2$ into the $HO_2$ signal.

RO$_2$ radicals are detected in the LIF instrument by a three step conversion of $RO_2$ to OH. Only species reacting with NO to RO and then decomposing or reacting with $O_2$ in a second reaction step





to $HO_2$ can be detected with a sufficient sensitivity. Depending on the model used up to 70 % of the modeled $RO_2$ species of $\beta$-pinene are not detectable under these conditions. To account for this, the

235 measured $RO_2$ signal $[RO_2^*]$ is compared to the model parameter $RO_2^*$, which corresponds to the sum of the theoretically detectable $RO_2$ model species.

The model $RO_2^*$ must be additionally corrected by the subtraction of the $RO_2$ species which are already included in the model parameter $HO_2^*$. This is again related to the operating conditions of the LIF instrument where in the $RO_x$ cell the sum of detectable $RO_2$ plus $HO_2$ and in the $HO_x$

cell $HO_2$ plus interfering $RO_2$ radicals are measured. As the $RO_2$ concentration is determined by subtracting the signal of the $HO_x$ cell from the signal of the $RO_x$ cell, an $RO_2$ interference in the $HO_x$ cell automatically leads to an underestimation of the calculated $RO_2$ concentration.

## 3 Results and discussion

### 3.1 Determination of product yields

The formation yields of first-generation degradation products are important information for the understanding of the oxidation mechanism of $\beta$-pinene with OH (Figure 5)). By correlating the concentration of the products with the concentration of the degraded $\beta$-pinene it is possible to determine the product yield. Because of the lack of suitable reference standards and the low concentration of $\beta$-pinene it was only possible to determine the yield of acetone and nopinone in the OH oxidation ex-

periment. The concentrations of $\beta$-pinene and nopinone were determined by PTR-TOF-MS whereas interpolated GC/FID data of the acetone concentration were used for the yield determination. This was done to exclude any possible interferences on the quantifier ion of acetone in the PTR-TOF-MS.

Caused by the addition of ozone in the experiment on 27 Aug 2012 a part of the injected $\beta$-pinene was degraded by ozonolysis. The fraction of the ozonolysis in the total conversion of $\beta$-pinene was

255 approximately 5 % and can be neglected.

The experiment duration of several hours necessitated the correction of the measured concentration time series to account for reactive losses of acetone and nopinone with OH and chamber effects like dilution (all species) and chamber sources (acetone). This was done using a recursive discrete time equation analogous to Galloway et al. (2011). The correction of the acetone concentration was

260 done by scaling the assumed acetone chamber source to the measured values during the zero air phase of the experiments. The assumed acetone source strength was typically $70 \, \mathrm{ppth^{-1}}$. Equations 3 - 7 illustrate all applied corrections on the acetone concentration.

$$[CH_3COCH_3]_{corr(i)} = [CH_3COCH_3]_{corr(i-1)} + \Delta c_{CH_3COCH_3} + \Delta c_{RL} + \Delta c_{DIL} + \Delta c_{S_{CH_3COCH_3}} \tag{3}$$

$$\Delta c_{RL} = [CH_3COCH_3]_{(i-1)} \cdot [OH]_{(i-1)} \cdot \Delta t \cdot k_{CH_3COCH_3+OH} \tag{4}$$





$$\Delta c_{DIL} = [CH_3COCH_3]_{(i-1)} \cdot \Delta t \cdot k_{DIL} \tag{5}$$

$$\Delta C_{S_{CH_3COCH_3}} = S_{CH_3COCH_3} \cdot \Delta t \tag{6}$$

$$S_{CH_3COCH_3} = a_{CH_3COCH_3} \cdot J_{NO_2} \cdot (0.21 + 2.6 \cdot 10^{-2} \cdot RH) \cdot e^{(-2876/T)} \tag{7}$$

$[CH_3COCH_3]_{corr}$: corrected acetone concentration

$\Delta c_{RL}$: reactive loss

$\Delta c_{DIL}$: dilution

$\Delta c_{S_{CH_3COCH_3}}$: chamber source

$\Delta t$: time interval between between time i and (i-1)

$S_{CH_3COCH_3}$: source strength

$a_{CH_3COCH_3}$: scaling factor

RH: relative humidity

$J_{NO_2}$: photolysis frequency $NO_2$

The results of the yield determination are summed up in Table 3. In principle product yields of nonlinear degradation processes depend on multiple physical and chemical boundary conditions as pressure, temperature, $H_2O$, $O_3$, VOC, $HO_2$ and NO concentration. The discussed $\beta$-pinene experiment was conducted at ambient pressure in a temperature range of $298 - 304\,K$. The relative humidity was about $50\,\%$ before the first VOC injection and decreased to $30\,\%$ over the course of the experiment, due to the warming of the chamber and the dilution of the chamber air by the replacement flow. It is known for many VOC species that the product yields depend on the VOC to NO ratio (Atkinson, 2000). This is why the two $\beta$-pinene experiments without and the $\beta$-pinene with the addition of $50\,ppb$ ozone are handled separately. During the experiment on 27th August the nopinone yield as well as the acetone yield increased subsequently with the second and third $\beta$-pinene addition and are therefore denoted as range. The specified errors consider the errors of measurement of the correlated VOC concentrations as well as the errors originating from the correction of reactive losses, dilution and chamber sources. To reduce the influence of secondary product formation and to facilitate the comparability of the results the yields were normalized to a conversion





of 70 % of the injected $\beta$-pinene. To our knowledge, these are the first acetone and nopinone yields measured for reaction mixtures with less than 5 ppb of $\beta$-pinene.

Within the calculated error the determined nopinone yield in this work agrees well with every literature value except the published yield of Hatakeyama et al. (1991). He is the only author reporting nopinone yields a factor of three higher then every other literature value. Vereecken and Peeters (2012) pointed out that Hatakeyama et al. (1991) measured the nopinone yield by using FTIR absorption at $1740\,\mathrm{cm}^{-1}$, which includes the absorption of other carbonyl compounds. Taking recent

literature and our results into account it seems that the nopinone yield of $\beta$-pinene oxidation with OH has no strong dependence on the NO level (see Table 3). The slight increase of the nopinone yield over the three $\beta$-pinene injections in the experiment of 27th Aug 2012 can be related to a change of boundary conditions as well as a secondary nopinone source. For example, the MCM 3.2 contains nopinone formation pathways from the degradation of the related hydroperoxides and

organic nitrates.

The determined acetone yield is in agreement with the reported literature values of Wisthaler et al. (2001), Librando and Tringali (2005) and Larsen et al. (2001). All reported literature values are smaller than the determined acetone yields in SAPHIR and show a wide range. Similar to nopinone there is no clear evidence of a NO dependence of the acetone yield. Due to the long reaction time the

increase of the acetone yield in the experiment of 27 Aug 2012 is most likely related to secondary acetone production. Since the yields in the literature were determined under various boundary conditions (e.g. light source, OH source, relative humidity) it is not possible to determine the reasons for the discrepancy. It could be related to different boundary conditions or measurement errors.

### 3.2    Comparison of the time series of trace gas concentrations with the MCM 3.2

In this section the measured time series of the $\beta$-pinene experiment from 27th August are compared to the base model using the unmodified MCM 3.2 (see Figure 1). The comparatively low VOC concentration allowed for the first time the intercomparison of modeled radical concentrations with parallel direct measurements of OH, $HO_2^*$ and $RO_2^*$. From the moment the roof of the SAPHIR chamber was opened HONO was formed at the chamber walls. Due to the photolysis of HONO OH

and NO were produced in the chamber, leading to a rise in the OH as well as the NO concentration. The parametrized HONO source sufficiently describes the measured nitrogen oxides in the zero air phase. The rise in the NO and $NO_2$ concentration is well captured. The modeled OH concentration also agreed well with the measurements.

Beside HONO also formaldehyde, acetaldehyde and acetone were formed or released from the

chamber walls, as can be seen in case of acetone in slight a concentration rise. These oxygenated VOC species (OVOCs) contributed to the increase of the measured background OH reactivity of $1.5\,\mathrm{s}^{-1}$ during the zero air phase of the experiment. As the sum of the measured OH reactants was not sufficient to explain the measured OH reactivity ($0.7\,\mathrm{s}^{-1}$ unexplained), the modeled OH reactivity





was adjusted by a constant source of a species Y, assumed to react like CO, i.e. with similar rate

coefficient and $HO_2$ formation. Under the assumption of a constant concentration of $120\,ppb$ Y the measured background reactivity is well reproduced by the model. Caused by the photochemical reactions of the detected OVOCs plus the unknown species contributing to the background reactivity $RO_2$ and $HO_2$ radicals are produced in SAPHIR, visible in a rise of the $RO_2^*$ and $HO_2^*$ concentration. The model underestimates the measured $RO_2^*$ concentration by $25\,\%$ whereas $HO_2^*$ is overestimated

by about a similar order of magnitude. In general the measured time series of radicals are less well captured by the MCM during the zero air phases of the experiments. The two main reasons for this are firstly that the OH reactivity is influenced by unknown species and secondly that radical sources and sinks are not well defined.

    With the beginning of the VOC phase of experiments, the OH reactivity is dominated by well-

known reactants, and much better model to measurement agreement is expected. For well-investigated reactants like CO and $CH_4$, agreements better than $15\,\%$ are typical for experiments in the SAPHIR chamber.

    For the current case, the addition of $\beta$-pinene led to a sharp increase in the measured OH reactivity. Directly after the $\beta$-pinene injection the increase of the modelled OH reactivity, calculated from

the canister injection, corresponded well with the measured $k(OH)$ increase. The $\beta$-pinene concentration measured by PTR-TOF-MS was about $15\,\%$ lower than then the calculated injection, but still agreed with the canister injection within the instrumental uncertainty. Over the course of the VOC phase, and thereby the consumption of $\beta$-pinene, the measured OH reactivity was increasingly overestimated by the model. During this time period nopinone has the highest proportion of modelled

OH reactivity beside $\beta$-pinene. However the measured nopinone concentration was overestimated by a factor of three by MCM 3.2 whereas the acetone concentration was underestimated by a factor of two. In general the MCM gives a poor description of the first-generation $\beta$-pinene degradation products. Simultaneously with the increase of the OH reactivity a sharp decrease of OH radical concentration was observed. At the time of $\beta$-pinene injection model and measurement agreed well,

but over the course of the experiment OH was increasingly underestimated by the model ($30$-$50\,\%$). The modeled concentration of theoretically measurable $RO_2$ radicals $RO_2^*$ exceeded the measured concentration by about $40\,\%$. Similar to OH, the modeled $HO_2^*$ concentration initially agreed well with the measurements directly after $\beta$-pinene injection but was increasingly underestimated by the MCM in the latter part of the experiment. The measured time series of ozone was well captured by

the MCM 3.2, whereas from the moment $\beta$-pinene was injected the model slightly overestimated the measured NO and $NO_2$ concentration

### 3.3 OH budget analysis

A complete model independent analysis of the radical chemistry taking place during the $\beta$-pinene oxidation is the analysis of the OH budget. In the OH budget analysis the measured OH production





rate $P_{OH}$ is compared to the measured OH destruction rate $D_{OH}$. Thereby $P_{OH}$ is defined as the sum

of all known OH production terms, in this case: the three primary OH production terms in SAPHIR

(HONO and $O_3$ photolysis, VOC ozonolysis) plus the OH production by the reaction of $HO_2$ with

$NO$ and $O_3$. Where $j_{O(^1D)}$ and $j_{HONO}$ are the measured photolysis frequencies of $O_3$ and HONO,

$f_{OH}$ is the fraction of $O(^1D)$ reacting with water to OH and $\alpha$ defines the OH yield of $\beta$-pinene

ozonolysis. The OH destruction $D_{OH}$ is given by the product of the measured OH reactivity and the

measured OH concentration. As $D_{OH}$ should be balanced by $P_{OH}$ during the whole experiment a

discrepancy in the budget gives a strong hint on possible missing OH production terms.

$$P_{OH} = j_{O(^1D)}[O_3] \cdot 2f_{OH} + j_{HONO}[HONO] + \alpha\, k_1[VOC][O_3] + k_2[HO_2][NO] + k_3[HO_2][O_3]$$
$$\text{(8)}$$

$$D_{OH} = k(OH) \cdot [OH] \tag{9}$$

$$P_{OH} = D_{OH} \tag{10}$$

Figure 2 displays the measured OH budget of the $\beta$-pinene experiment on 27th August 2012.

The lower panel of the plot shows the time series of the calculated OH turnover rates. The OH

destruction rate $D_{OH}$ is given as black line. The OH production rate $P_{OH}$ is shown by the sum of

the colored areas. Because of the higher instrumental accuracy the OH concentration measured by

the DOAS instrument was used to calculate $D_{OH}$. For $P_{OH}$ the OH recycling reaction of $HO_2$ with

$NO$ is the dominant OH production term followed by the photolysis of HONO. The OH production

by the ozonolysis reaction of $\beta$-pinene is of minor importance. As mentioned in the previous section

$HO_2$ measurements include an interference from specific $RO_2$. For the calculation of the measured

OH budget $HO_2$ data were not corrected for an $RO_2$ interference, as additional sensitivity studies

showed that the results of the budget analysis are not affected by an assumed $RO_2$ cross sensitivity of

25 %, because the derived $HO_2$ concentration would be lowered by less than 10 %. The upper panel

of figure 2 shows the time series of the ratio of $D_{OH}/P_{OH}$ (red line). The maximum systematic error

of $D_{OH}/P_{OH}$ is indicated by the grey area. Over the course of the experiment the measured OH

destruction rate is balanced by the sum of the quantifiable OH production terms within the maximum

systematic error as calculated from the sum of the uncertainties of the individual measurements.

Therefore the existence of large missing OH production terms can be excluded in the degradation

of $\beta$-pinene under the experimental conditions. The largest observed gap between $P_{OH}$ and $D_{OH}$

is in the order of 0.5 ppb/h. This result is different to previous studies conducted by Fuchs et al.

(2013, 2014) in the SAPHIR chamber, who, for isoprene and methacrolein, showed that significant

missing OH production terms could be identified in SAPHIR experiments. There, the measured OH

destruction rate was about a factor of two higher than the sum of the quantifiable OH production

terms. For the purpose of comparison the $\beta$-pinene study was conducted with the same experimental

setup and under comparable experimental conditions. The high relative uncertainty of $D_{OH}/P_{OH}$



during the zero air phase of the experiment is caused by the high uncertainty of the OH reactivity in that phase ( uncertainty $0.5\,\mathrm{s^{-1}}$ at a reactivity of $1\text{-}1.5\,\mathrm{s^{-1}}$). To assure the instrument data quality for the budget analysis the OH budget of well investigated reference systems (CO or $CH_4$ oxidation) was evaluated before and after the $\beta$-pinene campaign. Also in these experiments the measured OH destruction rate is balanced by the sum of the quantifiable OH production terms indicating that all

major OH production terms in SAPHIR are covered by the instrumental setup .

### 3.4 Modifications of the $\beta$-pinene oxidation mechanism

#### 3.4.1 Modified $\beta$-pinene oxidation mechanism by Vereecken and Peeters

As discussed in sections 3.1 (see table 3) and 3.2 the primary product yields of acetone and nopinone, calculated by the MCM 3.2, are not in agreement with the determined product yields un-

der low $NO_X$ conditions in SAPHIR as well as with yields reported in the literature. For further evaluation of radical chemistry processes a good reproduction of the first-generation $\beta$-pinene products is essential. An alternative model (Figure 5)) was published by Vereecken and Peeters (2012), including efforts to bring nopinone and acetone model yields in agreement with experimental data. Based on quantum chemical and theoretical kinetic calculations Vereecken and Peeters proposed a

fast ring opening reaction for the intermediate formed by the addition of OH to the double bond of $\beta$-pinene. This adjustment reduces the formation of the stabilized alkyl peroxy radical BPINAO2 (MCM specific designation), the main precursor in the MCM model for the nopinone formation, by about 70 %. Instead of BPINAO2, the $RO_2$ formed after ring opening, BPINCO2, is the dominant alkyl peroxy radical produced by the OH oxidation of $\beta$-pinene. With BPINCO2 as a starting

point Vereecken and Peeters developed a new degradation scheme for this branch of the $\beta$-pinene oxidation. In contrast to the original MCM 3.2 the primary acetone formation is now depending on two channels, leading to an increase of acetone formation under low $NO_X$ conditions, whereas the acetone yield in the MCM 3.2 is fairly stable. More details about the mechanism can be found in Vereecken and Peeters (2012). Their model was used without further changes except for the rate

constant of $\beta$-pinene with OH which was set to the MCM 3.2 value to facilitate model intercomparison. The original rate constant in the Vereecken and Peeters model refers to the published rate constant of Gill and Hites (2002) which is approximately 10 % lower. The result of the model calculation is shown in Figure 1 as blue line. In comparison to the MCM 3.2 the alternative $\beta$-pinene degradation scheme describes the measured time series of $k(OH)$ better, assuming $\beta$-pinene prod-

ucts with a lower OH reactivity. The time behaviour of the nopinone concentration is reproduced well by Vereecken and Peeters model. The acetone formation which was slightly underestimated by MCM 3.2 is now overestimated by nearly the same amount. It should be noted that the model by Vereecken and Peeters explicitly marks acetone formation in the current reaction conditions as a valuable metric to calibrate the acetone yield coming from a specific chemically-activated compe-





tition between different reaction channels available to alkoxy radical intermediate ROO6R2O. The current implementation assumes 100 % acetone formation; a more balanced value of 65 % would bring the acetone yield in agreement with the experiments

Table 4 further illustrates the difference of the product yields for acetone and nopinone calculated by the measured and modeled time series. To enable an intercomparison the product yields

calculated by modeled time series were also normalized to a $\beta$-pinene conversion of 70 %. All the corrections applied to the measured time series were applied in the same way to the modeled data. The measured nopinone yield of the first $\beta$-pinene injection is about 20 % lower then the nopinone yield observed for the 2nd and 3rd injection. This feature is well captured by the MCM model even if the total nopinone yield is approximately a factor of 2 too high. The reason for the increase in

the nopinone model yield is the secondary nopinone production by the degradation of previously formed hydroperoxides and organic nitrates originating from the same $RO_2$ radical which is also responsible for nopinone formation. In contrast to the MCM 3.2 the model of Vereecken and Peeters predicts a more stable nopinone yield. However, it does not include all secondary chemistry.

Over the three injections the measured acetone yield increased from 20 to 36 % , showing a clear

evidence for secondary acetone production. The MCM 3.2 as well as Vereecken and Peeters model also show an increasing acetone yield over time. In the MCM 3.2 the acetone yield is much too low compared to the measurements, but increases by a factor of three during the course of the experiment due to secondary acetone formations. The acetone yield calculated by Vereecken and Peeters model for the first injection is 70 % higher then the measured value. In contrast to the time behaviour of the

measured values the acetone yield is only slightly rising over the three injections, again possibly due to omitted secondary chemistry.

Concerning the agreement between measured and modeled radical concentrations the application of Vereecken and Peeters model does not lead to an improvement (see Figure 1). The measured OH and $HO_2^*$ concentrations are still underestimated in the VOC phase of the experiment. For $HO_2^*$

the decrease after the first $\beta$-pinene injection is even more pronounced. The reason for that is the $RO_2$ interference included in the modeled $HO_2^*$ data. In Vereecken and Peeters model less first-generation $RO_2$ radicals, formed by the oxidation of $\beta$-pinene by OH, can be theoretically detected by the LIF system. That's why directly after the first $\beta$-pinene injection the modeled observable $RO_2$ concentration by Vereecken and Peeters model is lower than in MCM 3.2. Simultaneously

this also means that the modeled $RO_2$ interference on the $HO_2^*$ time series is reduced. Compared to the measured time series of $RO_2^*$ Vereecken and Peeters model still overestimates the measured $RO_2^*$ concentration. Similar to the MCM 3.2 the measured NO and $NO_2$ concentration is slightly overestimated by the model, whereas the time series of ozone is captured well.

In summary therefore, it can be said that the alternative $\beta$-pinene degradation mechanism of

485 Vereecken and Peeters is able to describe the measured time series of nopinone, the measured OH reactivity and with that the OH losses during the experiment much better than the MCM 3.2. However,





these improvements do not lead to a satisfying description of the measured radical concentrations by the model, OH and $HO_2^*$ are still underestimated.

The good reproduction of the total OH loss together with the underestimation of OH and $HO_2^*$ by the model implies the need for an additional radical source to increase the modeled OH and $HO_2$ concentration. On the other hand the OH budget analysis clearly showed that the measurable OH sources were able to balance the measured total OH loss in the experiment. With this additional information of the previous OH budget analysis , indicating no significant missing OH source, there is the arising question how the radical production can be increased without overbalancing the OH budget. One option for that is the addition of an $HO_2$ source.

### 3.4.2 Oxidation mechanism by Vereecken and Peeters with measured $HO_2^*$ as model input

To investigate the influence of an additional $HO_2$ source on the modeled time series of all key species a primary $HO_2$ source was introduced in the model, taking the measured $HO_2^*$ data as model input. The known $RO_2$ interference in the measured $HO_2^*$ data was taken into account and corrected in the $HO_2$ model input. The result of the model run is displayed by the green curve in Figure 1. Applying an additional $HO_2$ source to the model improves the agreement of the modeled OH concentration with the measured values. In general the modeled OH increases by about 50 %. The higher OH level leads to an increase of chemical conversion over time, visible in a stronger decrease of $\beta$-pinene, nopinone and $k(OH)$ as well as in an increase of the modeled $RO_2^*$ concentration. Measured $\beta$-pinene, nopinone and $k(OH)$ are now underestimated by the model. A reason for that can be an underestimated $RO_2$ interference assumed for the $HO_2$ data, leading to a too strong $HO_2$ source in the model. In the case of the OH reactivity there is the additional uncertainty of the OH rate constants for the assumed $\beta$-pinene oxidation products beside nopinone, causing potentially a disagreement of modeled and measured $k(OH)$. For the overestimation of the measured $RO_2^*$ concentration one also has to take into account that the displayed time series of modeled $RO_2^*$ reflects the maximum $RO_2$ concentration which is theoretically detectable by LIF. An overestimation of the measured $RO_2^*$ concentration by the model might be related to an overestimation of the theoretically detectable $RO_2$ species in model or an incomplete conversion of $\beta$-pinene derived $RO_2$ radicals in the $RO_x$ cell of the LIF system. In addition the increase of the modeled $HO_2^*$ concentration leads to an improved description of the measured NO and $NO_2$ time series. Especially in the second half of the VOC phase the modeled NO and $NO_2$ concentration is reduced. As in any other model run there is no influence on the modeled ozone time series.

By the application of an $HO_2$ source to the model it was shown that the agreement between model and measurement could be improved for important key species like OH, NO and $NO_2$. Discrepancies in the OH lifetime and the $RO_2^*$ concentration could be attributed to uncertainties of the model. Therefore the evidence of a missing source of $HO_2$ in the degradation mechanism of $\beta$-pinene seems to be a reasonable hypothesis.



### 3.4.3 Uncertainties in the measured OH concentration

As stated in the previous section the input of the measured $HO_2$ concentration led to a satisfactory
description of the measured OH concentration by the model. On the other hand the elevated OH
concentration also resulted in an overestimated decrease of the $\beta$-pinene concentration measured by
PTR-TOF-MS. The reason for that is that the OH concentration calculated from the $\beta$-pinene decay
is about 31 % lower than measured by the LIF and 24 % lower than measured by the DOAS instru-
ment. Since both direct OH measurements agree well with each other and on the other hand the
decay of $\beta$-pinene measured by PTR-TOF-MS agrees well with the decay measured by GC/MS/FID
there is no clear indication for an instrumental failure or interference which would lead to an ex-
clusion of one or the other dataset. Because this contradiction cannot be solved the implications of
a potentially lower OH concentration on the previously discussed results should be elucidated. For
the OH budget analysis a 24 % lower OH concentration would lead to a decrease of the calculated
OH destruction $D_{OH}$ by an equal percentage. $D_{OH}$ would be overbalanced by $P_{OH}$ in a way that
the quotient $D_{OH}/ P_{OH}$ (0.76) would still be not significantly different from 1. As reported by Nehr
et al. (2014) for OH budgets during SAPHIR chamber experiments investigating CO as reference
system uncertainties of $\pm$ 20 % for $D_{OH}/ P_{OH}$ are common. For the comparison of the measured OH
concentration with the model calculations a 24 % lower measured OH concentration would result in
a reduced underestimation of the measured OH concentration by the models (now only 5-25 %)
whereas $HO_2^*$ would still be underestimated by a factor of two. Consequentially taking the corrected
$HO_2$ concentration as model input would result in an overestimation of the OH concentration by the
model up to 50 %. The influence of a 24 % lower measured OH concentration on the determined
product yields would be negligible because the corrections were small anyways.

### 3.4.4 Model studies to identify the reasons for the underestimation of $HO_2^*$

The model simulations in the previous section demonstrated that an unaccounted source of $HO_2$ is
a probable explanation for the disagreement of measured and modeled $HO_x$ concentrations. A com-
parison of the acquired results from the SAPHIR experiments with recent field campaigns shows
qualitatively the same results as in field studies which were conducted in forested areas dominated
by monoterpene emissions. Kim et al. (2013) reported a mismatch of the observed $HO_2$ concentra-
tion and model calculations. As in the SAPHIR experiments the OH budget was nearly balanced.
Kim et al. postulated a missing photolytic $HO_2$ source as the reason for the discrepancy between
the measured and modeled $HO_2$ concentration. Further investigations of the radical budget by Wolfe
et al. (2014) came to the same result. Additionally to the missing $HO_2$ source previously postu-
lated by Kim et al., Wolfe et al. also suggested a second peroxy radical source, being a photolytical
independent source of $RO_2$ radicals produced by the ozonolysis of unidentified VOC species. Sim-
ilar to Wolfe et al. and Kim et al. also Hens et al. (2014). reported that they found an unaccounted



primary $HO_2$ source, when they were comparing the measured time series of OH and $HO_2$ with model calculations. Under conditions of moderate observed OH reactivity and high actinic flux, an additional $RO_2$ source was needed to close the radical budget. Again also in the case of Hens et al. the measured OH budget was nearly balanced. In general it seems that the radical chemistry in a monoterpene dominated biogenic atmosphere in field campaigns or chamber studies recent atmospheric models underpredict the $HO_2$ production.

In atmospheric chemistry sources of $HO_2$ include the decomposition of alkoxy radicals, ozonolysis of VOCs and the reaction of CO, ozone or formaldehyde with OH, the photolysis of aldehydes and ketones and finally $HO_2$ formation by unimolecular rearrangement reactions of alkyl peroxy radicals (Orlando and Tyndall, 2012).

We investigate here two possible sources of $HO_2$ (see Figure 3): firstly, the formation of $HO_2$ by photolysis of reaction products, in particular aldehydes and ketones, and secondly by conversion of $RO_2$ to $HO_2$ without the involvement of NO. A third option to explain the discrepancy between modeled and measured $HO_2^*$ is the existence of an overestimated sink of hydroperoxyl radicals instead of a missing source of $HO_2$. A permanent sink of $HO_2$ in the $\beta$-pinene degradation mechanism is the formation of organic hydroperoxides (ROOH) by the reaction of $RO_2$ radicals with $HO_2$. Hence, the ROOH yield in the $RO_2$ reactions was varied in the third model run.

As the interference of $RO_2$ radicals in the measurements of $HO_2$ is also a subject of discussions, the maximum influence of the assumed $RO_2$ interference on the model results was estimated in a fourth model case (appendix A). The sensitivity study proved that the interference of the $RO_2$ radicals on the measured $HO_2$ time series is incapable to explain the observed deviations between modeled and measured $HO_2^*$. More than 50 % of the observed discrepancy cannot be explained by any known interference. For detailed information the reader is referred to additional information in the appendix.

For the addition of a primary photolytic $HO_2$ source an artificial species Z was introduced in the model. Every reaction of $\beta$-pinene with OH produces one molecule of Z additional to the related $RO_2$ species. Equivalent to an aldehyde the photolysis of Z, which is the only sink of the molecule beside dilution, produces an equal amount of $HO_2$ and CO. Not much is known about the photolysis of monoterpene degradation products. The only well investigated species is pinonaldehyde, a main degradation product of $\alpha$-pinene. As stated by Jaoui and Kamens (2003) the photolytic loss of pinonaldehyde is an important part of its overall atmospheric chemistry, accounting for as much as the loss by the reaction with OH during daytime. Nevertheless the photolysis rate of pinonaldehyde is very similar to the photolysis rate of formaldehyde. With regard to recent literature (Moortgat et al. (2002); Wenger (2006)), in general straight chain $C_3-C_9$ aldehydes and formaldehyde all have very similar photolysis rates. In contrast to that $\alpha$-branched aldehydes as well as aldehydes of substituted aromatics show significantly higher photolysis rates (factor of 3-20). As it is unlikely that degradation products of $\beta$-pinene form aromatic structures the photolysis rate of formaldehyde was





considered as a good first estimate for the photolysis rate of monoterpene degradation products in general. Therefore the photolysis rate of Z is set to the photolysis frequency of the radical pathway of the photolysis of formaldehyde. To increase the $HO_2$ source strength the number of $HO_2$ and CO molecules produced per photolysis of one molecule of Z was increased stepwise. The red curve shows the model run assuming a production of six $HO_2$ and CO molecules per molecule Z which

is in terms of chemical feasibility a rather unlikely assumption. Even if one takes into account the possibility of a photolytical cascade, a subsequent photolysis of the photolysis products, as proposed by Jaoui and Kamens (2003) for pinonaldehyde, a production of more than three molecules of $HO_2$ per molecule of $\beta$-pinene is rather implausible. Nevertheless under this assumption the modeled $HO_2^*$ agrees well with the measurements for the second half of the VOC phase. At the beginning

of the VOC phase $HO_2^*$ is still strongly underestimated because during that experimental phases there is still an insufficient amount of Z being built up yet (see Figure 3 red curves). Therefore the modeled OH concentration rises later than in the model run, taking measured $HO_2^*$ as input. Other modeled time series like $RO_2^*$, NO, $NO_2$, ozone and the $\beta$-pinene products acetone and nopinone stay nearly unchanged in comparison to the model run using measured $HO_2^*$ as model input. The

temporal behavior of modeled $HO_2^*$ and OH clearly shows that adding a photolytical $HO_2$ source cannot be a reasonable explanation for the missing $HO_2$ production in the model. Especially in the first half of the VOC phase the $HO_2$ production is too low to explain the measured $HO_2^*$ levels. Increasing the $HO_2$ production directly after the $\beta$-pinene injection would require a further increase in the photolysis rate to unrealistically high values. Furthermore, the assumption that one molecule

of Z has to produce six molecules of $HO_2$ and CO to match the measured $HO_2^*$ level at the end of the VOC phase demonstrates that a missing photolytical source cannot be the sole explanation for the disagreement between measured and modeled $HO_2^*$ in the second half of the VOC phase.

     In the second model run, displayed in blue in Figure 3, the impact of some chemical sources of $HO_2$ on the model are shown. To simulate the influence of unimolecular $RO_2$ rearrangements

without changing the whole degradation mechanism a modification of the so called X-mechanism published by Hofzumahaus et al. (2009) was used. An NO like species X is thereby reducing $RO_2$ radicals to RO radicals. The rate constants applied for these reactions are the same as the rate coefficients of NO with the corresponding $RO_2$ radical. Contrary to the X-mechanism of Hofzumahaus et al. in case of $\beta$-pinene X is not reacting with $HO_2$ radicals. For the model run, shown in Figure 3

a constant concentration of 300 ppt of X was assumed during the VOC phase. The comparison of the modeled $HO_2^*$ time series with the measurements shows that both time series are much better in agreement now, the measured $HO_2^*$ is underestimated less than 25 %. With the increase of modeled $HO_2^*$ also the modeled OH concentration increases and so agrees better with the measurements. In parallel the introduction of the new $RO_2$ loss pathway leads to a substantial decrease of the modeled

$RO_2^*$ concentration. The measured $RO_2^*$ is now slightly underestimated by the model. Furthermore the new $RO_2$ loss pathway has a strong influence on the modeled time series of NO and $NO_2$.





Both time series overestimate the measurements by up to 50 %. The reason for that is the reduction of organic nitrate formation. As the yield of organic nitrates is strongly depending on the carbon number (Koppmann, 2008) monoterpenes have relatively high nitrate yields, 18-26 % were reported for $\alpha$-pinene (Nozière et al., 1999; Rindelaub et al., 2014). If one adds fast NO independent $RO_2$ rearrangement reactions to the model these reactions compete with the NO reaction of $RO_2$ radicals and thus with the organic nitrate formation. With the X-mechanism the organic nitrate formation is reduced by about a factor of two compared to the base model of Vereecken and Peeters. Another effect is a slightly increased nopinone formation in the model. Again the comparison of the measured time series with the model shows the limited applicability of unimolecular $RO_2$ reactions acting as $HO_2$ source. In contrast to a photolytic source fast unimolecular rearrangement reactions are able to partially explain the measured $HO_2^*$ levels at the beginning of the VOC phase, but their influence on the organic nitrate yield leads to inconsistencies with the measured nitrogen oxide data.

The third sensitivity study (see the green curves in Figure 3) investigates the models response on a lower organic hydroperoxide yield. In the MCM 3.2 and the model of Vereecken and Peeters the ROOH formation is nearly exclusively determined by the rate constant KRO2HO2 which is applied to the majority of all $RO_2$ + $HO_2$ reactions. For the presented experiment KRO2HO2 was on average about 20.0 x $10^{-12}$ molecule$^{-1}$ s$^{-1}$. This value is in good agreement with rate coefficients for $RO_2$ + $HO_2$ reactions of hydroxy alkyl peroxy radicals generated from $\alpha$-pinene, d-limonene and $\gamma$-terpinene. Boyd et al. (2003) reported k-values of 19.7 -21.1 x $10^{-12}$ molecule$^{-1}$ s$^{-1}$ with an uncertainty of 7 - 20 %. For biogenic $RO_2$ species in general Orlando and Tyndall (2012) proposed an average uncertainty of a factor of two for the rate constants of $RO_2$ + $HO_2$ reactions. In the present sensitivity study the rate constant was therefore reduced by 50 %. As a result the modeled $HO_2^*$ concentration increases by 30 %, but $HO_2^*$ is still underestimated by the model. Similar to $HO_2^*$ also the modeled OH concentration slightly increases. The measured $RO_2^*$ concentration is overestimated by a factor of two by the model. A reduction of the organic hydroperoxide formation pathway also leads to a good model description of the measured time series of NO and $NO_2$. The reason for this is that a reduction of KRO2HO2 automatically yields in an increased production of organic nitrates which act as a temporary or permanent sink for nitrogen oxides. The influence of the reduced ROOH formation on other model species is small. Acetone formation is still over predicted by the model by 20 %, nopinone formation is slightly overpredicted as well as the $k$(OH) decay. The measured time series of $\beta$-pinene and ozone are reproduced well. A reduction of the ROOH production may help to reduce the discrepancy between the modeled and measured $HO_2^*$ concentration, but cannot solely explain the deviations between model and measurements.

Two model studies investigating the likelihood of two potential $HO_2$ sources to explain the missing $HO_2$ production term demonstrated that each source taken separately cannot explain the measured $HO_2^*$ level. Furthermore in the third study a reduction of the total hydroperoxide production rate by 50 % is not sufficient to explain the underestimation of the measured $HO_2^*$ concentration by



the model. A fourth study (see appendix A) investigating the influence of the $RO_2$ interference on
measured $HO_2^*$ time series also ruled out the assumption that the underestimation of $HO_2^*$ by the
model can be attributed to measurement interferences alone.

## 4 Summary and Conclusions

In this paper a set of three $\beta$-pinene oxidation experiments, conducted in the SAPHIR atmosphere
simulation chamber, was comprehensibly investigated with regard to the involved radical species
during the OH oxidation. A special focus was placed on the identification of possible missing OH
production terms in the degradation mechanism (Whalley et al., 2011). The experiments were con-
ducted under nearly ambient $\beta$-pinene concentration (4.3-4.7 ppb VOC) and low $NO_x$ conditions
(100-300 ppt NO). The comparatively low VOC concentration allowed for the first time the inves-
tigation of the radical budget of $\beta$-pinene by parallel measurements of OH, $HO_2$, $RO_2$ and $k$(OH).
In a first approach this comprehensive dataset was used for a model independent analysis of the
OH budget. For this purpose the sum of the measurable OH terms (HONO photolysis, $O_3$ pho-
tolysis, VOC ozonolysis, $HO_2$+NO, $HO_2$+$O_3$) was compared with the measured OH destruction
rate ($k$(OH) x [OH]). Contrary to previous studies of isoprene and methacrolein in SAPHIR (Fuchs
et al., 2013, 2014) the OH budget was balanced in the $\beta$-pinene oxidation experiments, giving no
evidence for significant missing OH production terms. In a second approach the measured time
series of the atmospheric key species were compared to 1-dimensional box model calculations to
investigate whether the models are able to predict the $\beta$-pinene degradation well. The comparison
of the measured time series with the MCM 3.2 revealed that the model was not able to reproduce
the measured time series of OH, $HO_2$, $k$(OH) and nopinone. The modeled OH as well as the $HO_2$
concentration was underestimated by more than 50 %. At the same time the modeled OH reactivity
was slightly overestimated. The reason for this disagreement is obviously a biased product distribu-
tion of the first-generation degradation products. The measured nopinone concentration was about a
factor of three lower than predicted by the model. A comparison of the experimentally determined
nopinone yield with recent literature showed a good agreement with recent literature but is a factor
of two lower than in the MCM model. Hence, for further investigations an updated MCM mecha-
nism published by Vereecken and Peeters (2012) was used. Their model was able to reproduce the
measured time series of nopinone and $k$(OH) much better than the MCM 3.2, but still significantly
underpredicted the measured OH and $HO_2$ concentration. As the previous analysis of the OH bud-
get showed no evidence of a missing OH source, an additional $HO_2$ source was introduced into the
model to improve the agreement for OH and $HO_2$. The sensitivity study showed that taking the mea-
sured $HO_2$ time series as model input generally improves the overall agreement of the modeled time
series with the measurements. OH is now well described by the model. These findings are qualita-
tively in agreement with recent field studies (Kim et al., 2013; Wolfe et al., 2014; Hens et al., 2014)





reporting that in a monoterpene dominated biogenic atmosphere models were not able to describe
OH and $HO_2$ levels well although the measured OH budget was balanced.

In accordance with the results for $\beta$-pinene presented in this paper we propose a missing photolytic
$HO_2$ source as the reason for the underestimation of OH in the model. An additional sensitivity study
trying to identify the nature of the $HO_2$ source for the $\beta$-pinene experiment showed that a formalde-
hyde like photolytic $HO_2$ source is not a reasonable option to explain the measured $HO_2$ and OH
levels. Due to the absence of a sufficient amount of photodegradable first generation products at
the beginning of the $\beta$-pinene oxidation a photolytic source is not able to produce enough $HO_2$ to
explain the measured concentration. A second sensitivity study demonstrated that the addition of
$RO_2$ rearrangement reactions releasing $HO_2$ to the model is not a reasonable option either. In case
of the $RO_2$ rearrangement reactions the added reaction path competes with the formation of organic
nitrates in the model and is thereby causing a strong overestimation of the measured nitrogen oxide
concentrations by the model. Additionally a third model run showed that an overestimated yield of
organic hydroperoxides can be excluded as the reason for the underestimation of the measured $HO_2$
concentration because the reduction of the $HO_2$ loss is too small. Further studies demonstrated that
an underestimation of the known $RO_2$ interference on the measurements of $HO_2$ can be excluded
as the reason for the observed high $HO_2$ concentrations. None of the previously discussed changes
in the mechanism as well as the $RO_2$ interference is able to solely explain the deviations between
model and measurements.

In conclusion, it can be said that the study of the $\beta$-pinene oxidation in SAPHIR as well as several
field campaigns showed the lack of understanding of the radical chemistry involved in the OH ox-
idation of monoterpenes. The identity of the proposed missing $HO_2$ source still remains uncertain.
To further elucidate the degradation mechanism for $\beta$-pinene and other monoterpenes more efforts
have to be made to quantify degradation products like organic nitrates, hydroperoxides, aldehydes
and ketones. Based on this knowledge proceeding investigations determining properties like photol-
ysis rates can be carried out. Potentially the yield of hydroperoxide formation, a sink for $HO_2$ can
also have an important influence on the modeled $HO_2$ and OH concentrations.

**Appendix A**

The interference of alkene-derived $RO_2$ radicals on the $HO_2$ measurements by LIF instruments
is a well known measurement uncertainty in the determination of the $HO_2$ concentration (Fuchs
et al., 2011; Whalley et al., 2013). Before the monoterpene measurement campaign the Jülich LIF
instrument was carefully characterized with regard to this interference. There average interference
of primary $RO_2$ radicals formed by the reaction of monoterpenes with OH was determined to be 25
$\pm$ 10 % on average. This interference was incorporated in the displayed model curves for $HO_2^*$ and
$RO_2^*$ in the modified MCM model (see the red curves). To exclude an impact of the $RO_2$ interfer-





ence on the proposed missing $HO_2$ source a sensitivity study was carried out. For that the previously
determined interference for primary $RO_2$s was doubled and applied on every $RO_2$ species in the
model. The result of this sensitivity study is displayed in the blue model curves in Figure 4 . As one
can see the enhanced $RO_2$ interference leads to 50 % increase in $HO_2^*$. But $HO_2^*$ is still underesti-
mated by 20-30 % in the model. Caused by the measurement principle of the LIF, determining the
$RO_2$ concentration by subtraction of the $HO_x$ from the $RO_x$ signal, an increase in the modeled $HO_2^*$
leads necessarily to a decrease in the modeled $RO_2^*$ concentration. In comparison to the base run
of the model of Vereecken and Peeters the modeled $RO_2^*$ level drops by 50 % and the model now
underestimates the measured $RO_2^*$ concentration.

*Acknowledgements.* This work was supported by the EU FP-7 program EUROCHAMP-2 (grant agreement
no. 228335) and by the EU FP-7 program PEGASOS (grant agreement no. 265307). S. Nehr and B. Bohn thank
the Deutsche Forschungsgemeinschaft for funding (grant BO 1580/3-1). The authors thank D. Klemp, C. Ehlers
and E. Kerger for provision of the EC/OC instrument for the determination of the VOC concentration in the
canisters and their technical support during the campaign. P. Schlag and D. F. Zhao are thanked for additional
particle phase measurements during this campaign. The service charges for this open access publication have
been covered by a Research Centre of the Helmholtz Association.





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



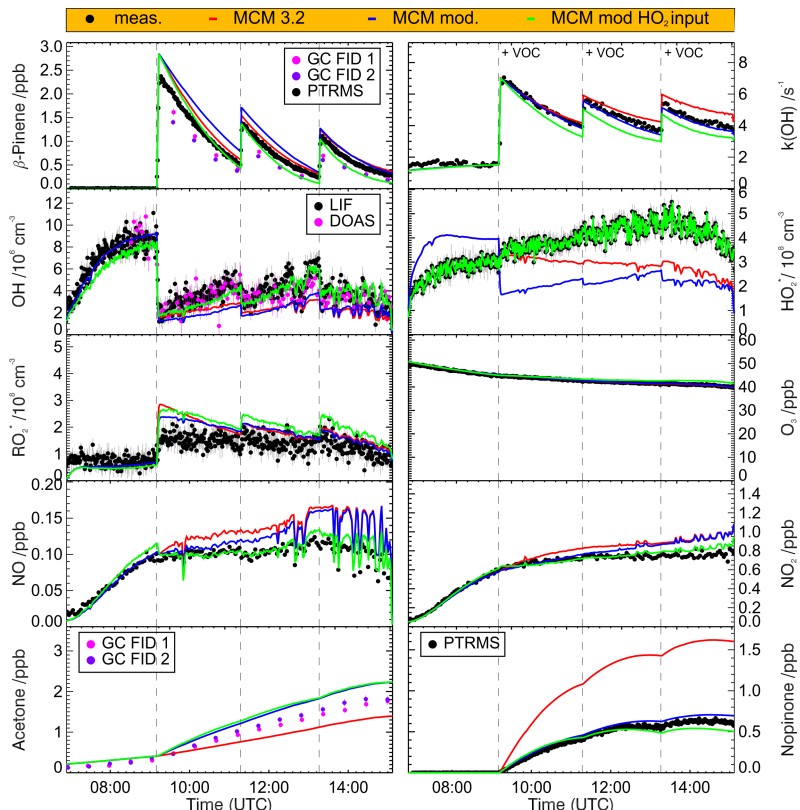

**Figure 1.** Comparison of the measured and modeled time series of $\beta$-pinene, $k$(OH), OH, HO$_2^*$, RO$_2^*$, NO, NO$_2$, acetone and nopinone in the $\beta$-pinene oxidation experiment from 27[th] August. Red: MCM 3.2 Blue: modified MCM model by Vereecken and Peeters (2012) with changed product yields Green: modified MCM model by Vereecken and Peeters (2012) constrained by the measured HO$_2$ concentration

(R1)




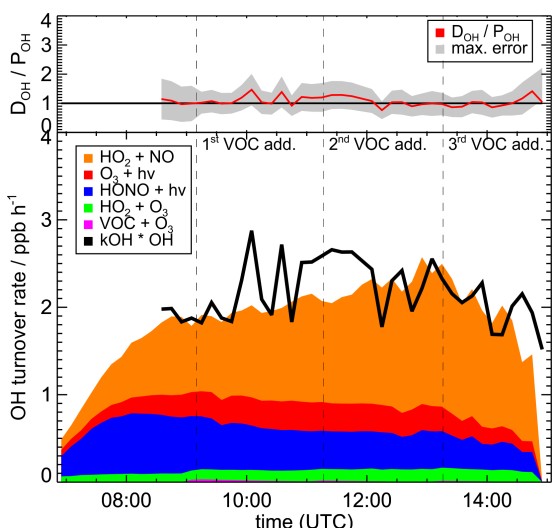

**Figure 2.** OH budget for the experiment on 27th August 2012. The OH destruction rate $D_{OH}$ calculated from the measured OH reactivity $k(OH)$ and the measured OH concentration (DOAS) is given as black line. The coloured areas display the OH production rate $P_{OH}$ calculated from measurements. The upper panel of the diagram shows the ratio of $D_{OH}/P_{OH}$ as red line. The maximum systematic error of the ratio is indicated by the grey area. For reasons of clarity all data in the upper as well as the lower panel of the diagram are shown as 5 min average values. During the course of the experiment the OH destruction rate is balanced by the sum of the measurable OH production terms. The reaction of $HO_2$ with NO and the photolysis of HONO are the dominant OH production terms. $HO_2$ measurements were not corrected for the interference from specific $RO_2$ species.



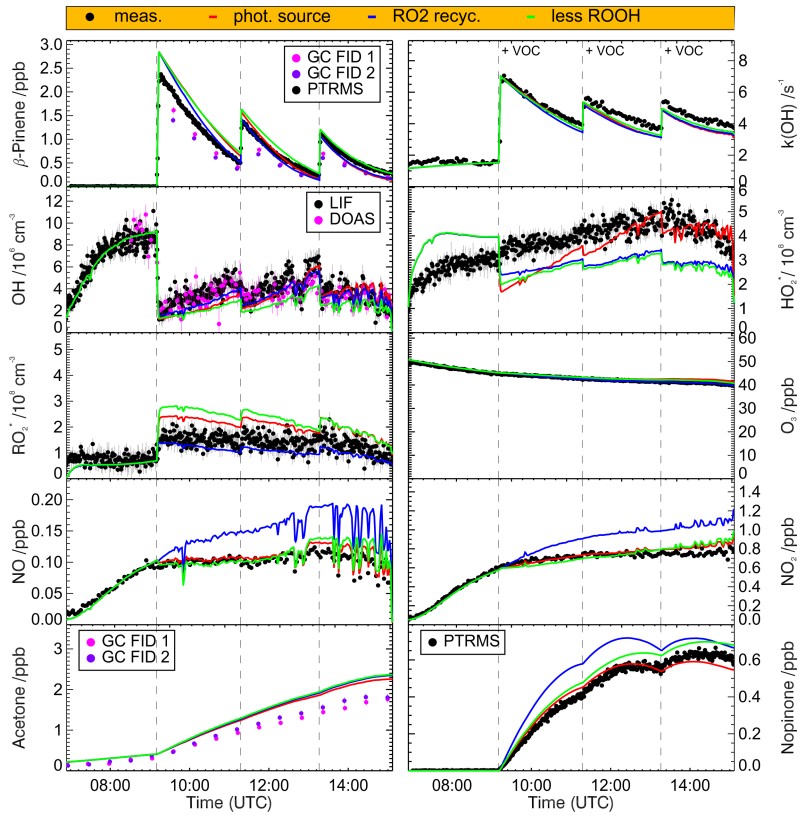

**Figure 3.** Comparison of the measured and modeled time series of $\beta$-pinene, $k$(OH), OH, $HO_2^*$, $RO_2^*$, NO, $NO_2$, acetone and nopinone in the $\beta$-pinene oxidation experiment from $27^{\text{th}}$ August. Red: modified MCM model by Vereecken and Peeters (2012) with additional photolytic source producing $HO_2$, Blue: modified MCM model with additional $HO_2$ formation by NO independent $RO_2$ rearrangement reactions, Green: modified MCM model with 50 % reduced yield of organic hydroperoxides ROOH



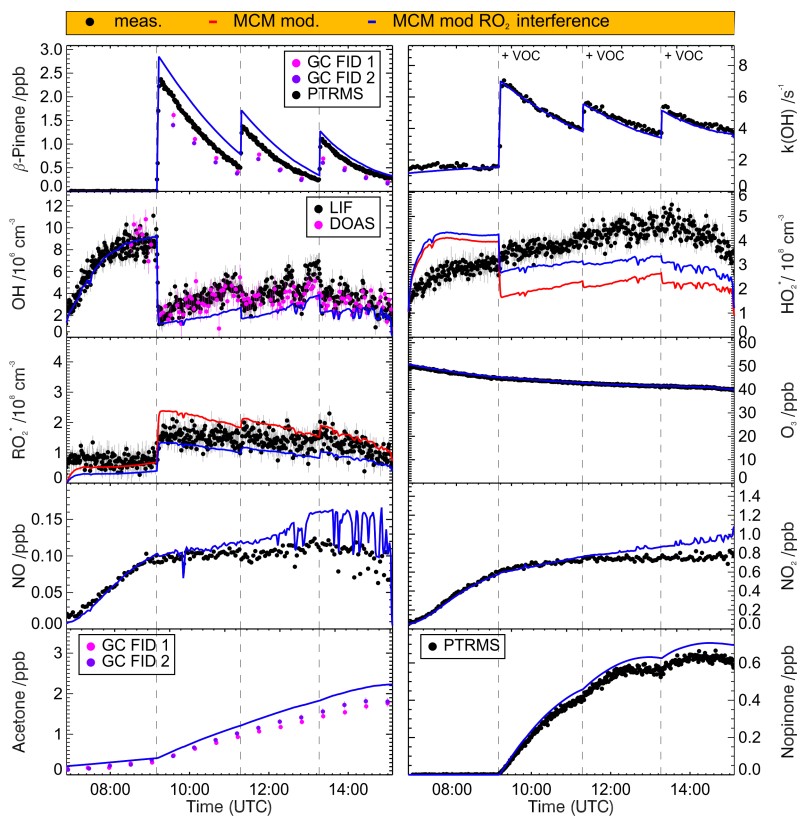

**Figure 4.** Comparison of the measured and modeled time series of $\beta$-pinene, $k(OH)$, OH, $HO_2^*$, $RO_2^*$, NO, $NO_2$, acetone and nopinone in the $\beta$-pinene oxidation experiment from 27[th] August. Red: modified MCM model by Vereecken and Peeters (2012) with changed product yields, Blue: modified MCM model assuming an $RO_2$ interference of 50 % for the $HO_2$ measurements





**Figure 5.** Acetone and nopinone formation from OH initiated $\beta$-pinene oxidation after Vereecken and Peeters

(2012). For simplification only the major reactions are shown.





**Table 1.** Instrumentation for radical and trace gas detection during the $\beta$-pinene oxidation experiments.

| | Technique | Time Resolution | $1\sigma$ Precision | $1\sigma$ Accuracy |
|---|---|---|---|---|
| OH | DOAS[a] (Dorn et al., 1995; Hausmann et al., 1997; Schlosser et al., 2007) | 205 s | $0.8 \times 10^6\,\mathrm{cm^{-3}}$ | 6.5 % |
| OH | LIF[b] (Lu et al., 2012) | 47 s | $0.3 \times 10^6\,\mathrm{cm^{-3}}$ | 13 % |
| $HO_2$, $RO_2$ | LIF[b] (Fuchs et al., 2011) | 47 s | $1.5 \times 10^7\,\mathrm{cm^{-3}}$ | 16 % |
| $k(OH)$ | Laser-photolysis + LIF[b] (Lou et al., 2010) | 180 s | $0.3\,\mathrm{s^{-1}}$ | $0.5\,\mathrm{s^{-1}}$ |
| NO | Chemiluminescence (Rohrer and Brüning, 1992) | 180 s | 4 pptv | 5 % |
| $NO_2$ | Chemiluminescence (Rohrer and Brüning, 1992) | 180 s | 2 pptv | 5 % |
| $O_3$ | Chemiluminescence (Ridley et al., 1992) | 180 s | 60 pptv | 5 % |
| VOCs | PTR-TOF-MS[c] (Lindinger et al., 1998; Jordan et al., 2009) | 30 s | 15 pptv | 14 % |
| | GC[e] (Kaminski, 2014) | 30 min | 4-8 % | 5 % |
| CO | RGA[f] (Wegener et al., 2007) | 3 min | 4 % | 10 % |
| HONO | LOPAP[d] (Häseler et al., 2009) | 300 s | 1.3 pptv | 10 % |
| HCHO | Hantzsch monitor (Kelly and Fortune, 1994) | 120 s | 20 pptv | 5 % |
| Photolysis frequencies | Spectroradiometer (Bohn and Zilken, 2005) | 60 s | 10 % | 10 % |

[a] Differential Optical Absorption Spectroscopy.
[b] Laser Induced Fluorescence.
[c] Proton-Transfer-Reaction Time-Of-Flight Mass-Spectrometry.
[d] LOng Path Absorption Photometer.
[e] Gas Chromatography.
[e] Reactive Gas Analyzer.

**Table 2.** Experimental conditions of the $\beta$-pinene oxidation experiments. Maximum values are given for $\beta$-pinene and averaged values for the part of the experiment, when $\beta$-pinene was present, for the other parameters.

| $\beta$-pinene ppbv | OH $10^6\,\mathrm{cm^{-3}}$ | $NO_x$ ppbv | NO pptv | $O_3$ ppbv | RH % | $j(NO_2)$ $10^{-3}\,\mathrm{s^{-1}}$ | $T$ K | date |
|---|---|---|---|---|---|---|---|---|
| 4.3 | 6.0 | 1.0 | 300 | 10 | 45 | 5 | 295 | 12 Aug 2012 |
| 4.3 | 4.5 | 0.9 | 200 | 10 | 45 | 4 | 299 | 15 Aug 2012 |
| 4.7 | 3.5 | 0.9 | 100 | 40 | 40 | 4.5 | 293 | 27 Aug 2012 |



**Table 3.** Product yields from the reaction of $\beta$-pinene with OH radicals under various NO and VOC concentrations

| Product | Yield OH reaction | Reference | consumed VOC ppbv | NO ppbv |
|---|---|---|---|---|
| Nopinone | 0.35 ±0.13 | This work | 3 | 0.4 |
| | 0.28-0.37 ±0.13 | | 3 | 0.1 |
| | 0.79[a] ±0.08 | Hatakeyama et al. (1991) | 700 | 1800 |
| | 0.30 ±0.045 | Arey et al. (1990) | 960 | 960 |
| | 0.27 ±0.04 | Hakola et al. (1994) | 1000 | 9600 |
| | 0.25 ±0.05 | Larsen et al. (2001) | 1300-1600 | 0 |
| | 0.25 ±0.03 | Wisthaler et al. (2001) | 1000-3000 | 1000-2000 |
| | 0.24 | Librando and Tringali (2005) | 4100-13200 | 0 |
| Acetone | 0.19 ±0.06 | This work | 3 | 0.4 |
| | 0.20-36 ±0.07 | | 3 | 0.1 |
| | 0.13 ±0.02 | Wisthaler et al. (2001) | 1000-3000 | 1000-2000 |
| | 0.11 ±0.03 | Larsen et al. (2001) | 1300-1600 | 0 |
| | 0.03-0.06 | Fantechi (1999) | | |
| | 0.02 ±0.002 | Orlando et al. (2000) | 1800-12000 | 800-8000 |
| | 0.085 ±0.018 | Reissell et al. (1999) | 880-920 | 9600 |
| | 0.14 | Librando and Tringali (2005) | 4100-13200 | 0 |

[a] Yield measured by FTIR absorption at 1740 cm-1, and is suspected to include other carbonyl compounds

**Table 4.** Comparison of measured and modeled product yields from the reaction of $\beta$-pinene with OH radicals for the three $\beta$-pinene injections during the experiment on 27 th August 2012

| Product | Injection | Yield measured | Yield MCM 3.2 | Yield Vereecken and Peeters |
|---|---|---|---|---|
| Nopinone | 1 st | 0.28 | 0.53 | 0.27 |
| | 2 nd | 0.37 | 0.61 | 0.28 |
| | 3 rd | 0.35 | 0.65 | 0.30 |
| Acetone | 1 st | 0.20 | 0.07 | 0.37 |
| | 2 nd | 0.24 | 0.16 | 0.47 |
| | 3 rd | 0.36 | 0.21 | 0.49 |