# Peer review of "Investigation of the $\beta$ -pinene photooxidation by OH in the atmosphere simulation chamber SAPHIR"

_Atmospheric Chemistry and Physics, 2016_

## Referee Comment (RC1) · Anonymous Referee #1 · 17 Dec 2016

Review of "Investigation of the B-pinene photooxidation by OH in the atmosphere simulation chamber SAPHIR", Kaminski et al., ACP (2016)

**Summary**

This paper analyzes a set of chamber experiments focused on the chemistry of OH + B-pinene. Unlike previous studies, these experiments are done at relatively low VOC and NOx loading, thus better representing true atmospheric conditions. The chamber is also highly instrumented, allowing examination of both radical chemistry and product formation. Interpretation is aided by a box model. The authors find that formation of first-generation products is represented well by a recent theory-based mechanism, but not the MCM. They further identify a major discrepancy between modeled and observed HO2, which they ascribe to an unknown radical source.

The results of these experiments are interesting, novel, and worthy of publication in ACP. The English is verbose/awkward in places, but still understandable. The text is a little long and could be shortened in places. My primary concerns lie in the interpretation of the radical budgets, and in particular the potential for chamber or measurement-based artifacts. Publication is recommended after considering the following.

**General Comments**

The reviewer is not convinced that a missing HO2 source is the most reasonable hypothesis to fit the observations. Two specific concerns in this regard:

1) The decay of B-pinene is well described by the base model simulation, and any attempts to artificially increase model HOx degrade this agreement. This is discussed in Section 3.4.3, where it is stated that two independent measurements of both OH and B-pinene (decay) agree well with one another, therefore there seems to be no obvious explanation. One could argue, however, that observations of absolute OH concentration are susceptible to systematic errors, while the decay rate of a VOC is less likely to have such errors. Indeed, the observed decay rate of B-Pinene coupled with the uncertainty in the OH+BPIN rate constant should bracket the range of "reasonable" OH concentrations. The alternative is that an additional process is causing the B-Pinene decay to be artificially slow, but no explanation is given to that end.

2) HO2* in Fig. 1 climbs steadily throughout the experiment and reaches a maximum around 1300 UTC; indeed, the trend before and after B-pinene injection is fairly similar. No other species show this behavior. My concern is that this is a chamber wall or instrument artifact, related to either the build-up of oxidation products and/or insolation. It would be nice to see, for example, what the time-progression of J(HCHO) is over this experiment. Does it also show a maximum at ~1300 UTC? The mystery HO2* source reported by Wolfe et al. (2014) showed a marked dependence on solar radiation, so this would provide some support that the two phenomena (whatever their origin) are similar. If J(HCHO) does show

a similar trend to HO2*, could the "photolytic source" from molecule "Z" match HO2* if it were set to a constant value instead of being tied to B-pinene oxidation? Perhaps some long-lived contaminant is injected along with B-Pinene.

**Specific Comments**

L10: when referring to "low NOx" in this context, it is better to give NO than NOx mixing ratios since this determines the radical fate.

L18: Factor of 3 low or high?

L55: It should also be mentioned that we have since discovered substantial artifacts in some HOx instruments for some conditions (Mao et al., 2012).

L190: Do these source strengths vary significantly day-to-day? Just curious if this could indicate something about chamber wall aging.

L246: Suggest ensuring figures and tables are numbered to reflect the order they appear in the text (this may even by ACP policy).

L261: How does this 70 ppt/h compare to production rate of acetone from BPIN? From figures , it looks like it is relatively small, but it would add confidence to say so.

L296: More generally, one could just say that the product yield depends on the fate of RO2.

L307: "the yields were normalized to a conversion of 70% of the injected B-pinene." I do not understand what this means; please clarify.

L316: Do either the MCM or Vereecken and Peeters mechanisms exhibit an NO dependence in the yield? This could be examined fairly easily with the box model, ramping NO between say 10 ppt and 1 ppb.

L418: In these experiments, do measured and modeled HO2* agree well?

L438: What is meant by "stable?"

Section 3.4.4: This section could be shortened significantly by removing details. It seems like a lot of text to present 3 sensitivity studies, none of which explain the missing HO2.

L555: Some caution is warranted when comparing to these field studies, as the VOC were quite different. For example, in the works of Kim and Wolfe, MBO was the dominant VOC.

L600: Z is produced from B-pinene oxidation with an arbitrary yield of 1, so the choice of an equally arbitrary HO2 yield of 6 doesn't seem very unreasonable. There may also be multiple generations of chemistry that are being wrapped up into Z.

Section 4: First paragraph could be shortened a lot. Don't need to re-outline the whole paper, just highlight the key findings. Also in last paragraph, please be more specific about what types of experiments, or what measurements, are needed to pin down this missing HO2 source. SAPHIR is already pretty well-armed.

**Technical Comments**

L35: delete "to OH"

L180: delete "are"

L221: "interfering"

L436: "formation now depends"

L674: "comprehensively"

Figure 1: suggest changing the name of the "MCM mod" run to something like "VP2012." If I read the text right, this is not MCM at all (except for maybe the inorganic chemistry).

Figure 2: Suggest setting the y scale for the top plot to 0-2.

Figure 3: Please include a line for the base simulation for comparison.

**References**

Mao, J., Ren, X., Brune, W. H., Van Duin, D. M., Cohen, R. C., Park, J. H., Goldstein, A. H., Paulot, F., Beaver, M. R., Crounse, J. D., Wennberg, P. O., DiGangi, J. P., Henry, S. B., Keutsch, F. N., Park, C., Schade, G. W., Wolfe, G. M., and Thornton, J. A.: Insights into hydroxyl measurements and atmospheric oxidation in a California forest, Atmos. Chem. Phys., 12, 8009-8020, doi: 10.5194/acp-12-8009-2012, 2012.

---

## Referee Comment (RC2) · Anonymous Referee #2 · 23 Dec 2016

This paper describes experiments conducted in the SAPHIR chamber in Julich designed to study the photooxidation of beta-pinene. The chamber is well equipped with instrumentation to measure both free radicals (OH, HO2 and RO2) and stable molecules. Consequently, the study focuses mostly on the radical budget, and investigates whether the OH and HO2 rates of production and loss can be determined.

Experimentally, the measured rates of production and loss of OH are found to balance, in contrast to previous studies of isoprene and methacrolein chamber oxidation. However, a model analysis found lower rates of production and destruction, and overall lower radical concentrations. Use of a more detailed model, and measured HO2 concentrations, partially reduced the discrepancy, and sensitivity analyses showed that

the measured data could be better represented by introducing an unidentified source of HO2.

Overall, this is a good paper. The experiments and modeling are well described, and attention is paid to uncertainties in the system.

I was a little surprised that only one experiment of the three was analysed in any detail. In fact, no data were shown from two of the experiments (just briefly in the Table). I think this is a serious omission, as these were "normal" experiments, with no added O3, and no extra additions of B-pinene. Would it be possible to include some of these experiments to contrast the results? A wider variation of the NO concentration would be useful. Also, it is unfortunate that no product information is given other than for acetone and nopinone, despite the fact that a PTRMS was used for analysis.

Other Comments Line 48. I think it would be better to say that results showed "an incomplete knowledge" rather than "a lack of knowledge".

Line 104 (and elsewhere). Insert "such" before "as" i.e. "species such as. . ."

Line 115 Change "effects" to "affects"

Line 167. MCM is a zero-D, or box model (1-D usually refers to a column model with vertical transport).

Line 180. Delete "are".

Line 221. Inferring should be Interfering?

Line 246. Delete one of the double parentheses.

Line 253. Change "Caused by" to "As a result of. . ."

Line 295. Again, insert "such" before "as".

Line 312. Change "then" to "than".

Line 346. Clumsy sentence beginning "Caused by. . .".

Line 425. "under low NOx conditions". I realize this is somewhat a matter of semantics, and under much discussion at present, but be careful how you classify the NOx environment. With 100 ppt of NO (measured) and up to 20 ppt of HO2, >50% of the RO2 radicals will still react with NO. So it is not strictly a low-NOx environment.

Line 425 or so. I know Vereecken and Peeters ruled this out on the basis of barrier heights, but could a chemically activated BPINAO radical decompose by ring opening, rather than formation of nopinone and CH2OH? It might help to explain the acetone/nopinone dilemma.

Line 427. Remove double parenthesis.

Lines 436-438, and 4447-450. I think a few more words of clarification might be useful here for people not familiar with the mechanism.

As I understand it, the original (as in MCM) fate of BPINCO2 was to react with NO to make acetone predominantly. In the Vereecken and Peeters mechanism, this radical can isomerise under low NOx (to make a bicyclic peroxy radical, which then reacts with NO to make acetone, via a chemically activated alkoxy radical. However, this alkoxy radical can also isomerise (from the aldehyde group) to give different products. A few words describing this train of thought would be helpful. Particularly, be more specific about the radicals involved and how they are reacting.

So why should the acetone yield increase at low NOx (line 437), if the MCM predicts only acetone as a product?

Again, a more detailed analysis of some experiments with varying NO concentrations would have been very useful to diagnose this.

Lines 596-600. As the authors agree, such a large source of HO2 from photolysis of a carbonyl is implausible. But is it? Presumably photolysis leads to the production of 2 radicals. So increasing the photolysis rate by a factor of 3 would work, rather than producing 6 radicals. Is it possible that it is a dicarbonyl similar to glyoxal, which have

very fast photolysis rates?

Line 686. 1-dimensional should be zero-dimensional.

---

## Author Comment (AC1) · 6 Apr 2017

We thank the reviewer for his/her comments. Here are our responses to the specific comments.

Review of "Investigation of the B-pinene photooxidation by OH in the atmosphere simulation chamber SAPHIR", Kaminski et al., ACP (2016)

Summary

This paper analyzes a set of chamber experiments focused on the chemistry of OH + B-pinene. Unlike previous studies, these experiments are done at relatively low VOC and NOx loading, thus better representing true atmospheric conditions. The chamber

is also highly instrumented, allowing examination of both radical chemistry and product formation. Interpretation is aided by a box model. The authors find that formation of first-generation products is represented well by a recent theory-based mechanism, but not the MCM. They further identify a major discrepancy between modeled and observed HO2, which they ascribe to an unknown radical source. The results of these experiments are interesting, novel, and worthy of publication in ACP. The English is verbose/awkward in places, but still understandable. The text is a little long and could be shortened in places. My primary concerns lie in the interpretation of the radical budgets, and in particular the potential for chamber or measurement-based artifacts. Publication is recommended after considering the following.

General Comments

The reviewer is not convinced that a missing HO2 source is the most reasonable hypothesis to fit the observations. Two specific concerns in this regard:

Comment

1) The decay of B-pinene is well described by the base model simulation, and any attempts to artificially increase model HOx degrade this agreement. This is discussed in Section 3.4.3, where it is stated that two independent measurements of both OH and B-pinene (decay) agree well with one another, therefore there seems to be no obvious explanation. One could argue, however, that observations of absolute OH concentration are susceptible to systematic errors, while the decay rate of a VOC is less likely to have such errors. Indeed, the observed decay rate of B-Pinene coupled with the uncertainty in the OH+BPIN rate constant should bracket the range of "reasonable" OH concentrations. The alternative is that an additional process is causing the B-Pinene decay to be artificially slow, but no explanation is given to that end.

Response

As explained in chapter 3.4.3 there is a discrepancy between the OH concentration

measured directly by LIF and DOAS and the OH concentration as calculated from the measured decay of b-pinene which we cannot explain. However, with any OH dataset the OH production rate corresponds the measured OH destruction rate with in the uncertainty of the instruments. The OH budget is closed. Also, when the measured HO2 concentration in the model is constraint with measured HO2 data, the difference between measured and modelled OH is small. For most of the experiment, the modelled OH concentration is higher than the OH concentration inferred from the VOC measurements and lower than the directly measured data. Therefore, the discrepancy of OH data from direct measurements and OH data inferred from the VOC data is not of importance for the key findings of the paper.

Comment

2) HO2* in Fig. 1 climbs steadily throughout the experiment and reaches a maximum around 1300 UTC; indeed, the trend before and after b-pinene injection is fairly similar. No other species show this behavior. My concern is that this is a chamber wall or instrument artifact, related to either the build-up of oxidation products and/or insolation. It would be nice to see, for example, what the time-progression of J(HCHO) is over this experiment. Does it also show a maximum at ∼1300 UTC? The mystery HO2* source reported by Wolfe et al. (2014) showed a marked dependence on solar radiation, so this would provide some support that the two phenomena (whatever their origin) are similar. If J(HCHO) does show a similar trend to HO2*, could the "photolytic source" from molecule "Z" match HO2* if it were set to a constant value instead of being tied to b -pinene oxidation? Perhaps some long-lived contaminant is injected along with b -Pinene.

Response

The interference of the HO2 data for RO2 radicals has been determined prior to the monoterpene degradation campaign in lab experiments. A sensitivity study (see supporting material) showed that even a doubling the interference would not be enough to

explain the measured HO2 values. We agree that the measured HO2* values do not change significantly before and after the VOC injections. A similar time series of HO2* was observed in all b -pinene experiments, but not in limonene and a-pinene experiments which were conducted in the same time period. Therefore this HO2 time series is chemistry related and not caused by artefacts of the instrument. The measured photolysis frequency of formaldehyde J(HCHO) had maximum values between 10:00 AM and 1:30 PM as the measured HO2, but this would be expected from the model, if a photolytic source of HO2 is assumed: In standard experiments HO2* agrees within 15% with the modelled values. Also we did not observed indications for impurities being injected together with the b-pinene. The compound was specified with a purity of 99%, and contaminants were detected neither by PTRMS nor by GC/MS.

Comment

L10: when referring to "low NOx" in this context, it is better to give NO than NOx mixing ratios since this determines the radical fate.

Response

NOx is changed to NO in the text

Comment

L18: Factor of 3 low or high?

Response

Nopinone concentration is three times higher than measured. The sentence is rephrased.

Changed text:

Old: The measured OH and HO2 concentrations were underestimated by up to a factor of two whereas the total OH reactivity was slightly overestimated because of the poor reproduction of the measured nopinone by the model by up to a factor of three

New: The measured OH and HO2 concentrations were underestimated by up to a factor of two whereas the total OH reactivity was slightly overestimated because the model predicted a nopinone mixing ratio which was three times higher than measured.

Comment

L55: It should also be mentioned that we have since discovered substantial artifacts in some HOx instruments for some conditions (Mao et al., 2012).

Response

The study is added to the text.

Changed text:

The newly discovered mechanisms for isoprene and methacrolein, however, can explain only part of the observed high OH concentrations. Another possible reason could be OH interferences in the low pressure laser-induced fluorescence (LIF) instruments that were applied in the above field studies. Artificial OH production was discovered in two similar LIF instruments applying a newly developed chemical modulation technique for OH detection (Mao et al., 2012; Hens et al., 2014; 5 Novelli et al., 2014; Feiner et al., 2016). The interference seems to be related to organic compounds, but the underlying OH formation mechanism is not known. Experimental tests with other type of LIF instruments have not found such interference (Fuchs et al., 2012, 2016; Griffith et al., 2013; Tan et al., 2017), yet it is difficult to draw firm conclusions for past campaigns as long as the reported artefacts (Mao et al., 2012) are not fully understood.

Comment

L190: Do these source strengths vary significantly day-to-day? Just curious if this could indicate something about chamber wall aging.

Response

The chamber strength of HONO, formaldehyde and acetone can be calculated with

good precision from the measured photolysis frequency of NO2 (JNO2), the relative humidity and the temperature using a scaling factor. Prior to each experiment a zero experiment was conducted to determine the scaling factor needed to explain the observed values of these trace compounds. While both the scaling factors for the HONO source and the acetone source did not vary by more than 20% from day to day over the one month period, the scaling factor for formaldehyde increased by a factor of three throughout the campaign.

Comment

L246: Suggest ensuring figures and tables are numbered to reflect the order they appear in the text (this may even by ACP policy).

Response

Figure numbering is changed.

Comment

L261: How does this 70 ppt/h compare to production rate of acetone from BPIN? From figures, it looks like it is relatively small, but it would add confidence to say so.

Response

20% to 30% of the total acetone formed in the b-pinene experiment was due to emissions production of the chamber wall.

Changed text:

Old: The assumed acetone source strength was typically 70 ppth-1.

New: The assumed acetone source strength was typically 70 ppth-1 which was as large as 20 to 30 % of the total amount of acetone produced in the ß-pinene experiments.

Comment

L296: More generally, one could just say that the product yield depends on the fate of

[Figure]

RO2.

Response

This has been changed in the text

Changed text:

Old: In principle product yields of nonlinear degradation processes depend on multiple physical and chemical boundary conditions as pressure, temperature, H2O, O3 , VOC, HO2 and NO concentration.

New: In principle product yields of nonlinear degradation processes depend on the fate of RO2 which is governed by multiple physical and chemical boundary conditions such as pressure, temperature, H2O, O3 , VOC, HO2 and NO concentration.

Comment

L307: "the yields were normalized to a conversion of 70% of the injected B-pinene." I do not understand what this means; please clarify.

Response

The product yields were determined from a linear fit of b-pinene consumed versus the amount of acetone produced. During the course of the experiment the yield of acetone increases due to the increased production from secondary products. Therefore, only the data of experiments were used for the yield determination when less than 70% of b-pinene was consumed, i.e. when the chemistry of secondary products is still minor relative to the b-pinene chemistry.

Comment

L316: Do either the MCM or Vereecken and Peeters mechanisms exhibit an NO dependence in the yield? This could be examined fairly easily with the box model, ramping NO between say 10 ppt and 1 ppb.

Response

As proposed, NO was ramped in the model from 10 to 1000 ppt NO. The MCM model suggests an increase from 0.44 at 10 ppt to 0.7 at 1 ppb. The Vereecken model predicts only an increase from 0.21 at 10 ppt NO to 0.26 at 1 ppb.

Comment

L418: In these experiments, do measured and modeled HO2* agree well?

Response

In standard experiments HO2* agrees within 15% with the modelled values.

Comment

L438: What is meant by "stable?"

Response

The sentence has been changed.

Changed text:

Old: In contrast to the original MCM 3.2 the primary acetone formation is now depending on two channels, leading to an increase of acetone formation under low NOX conditions, whereas the acetone yield in the MCM 3.2 is fairly stable. More details about the mechanism can be found in Vereecken and Peeters (2012).

New: This leads to an increase of acetone formation at low NO concentrations compared to the MCM 3.2 while the yield of nopinone is predicted to be lower in the model by Vereecken and Peeters (2012).

Comment

Section 3.4.4: This section could be shortened significantly by removing details. It seems like a lot of text to present 3 sensitivity studies, none of which explain the missing

HO2.

Response

This section has been shortened. The sensitivity studies are only mentioned now. The detailed description of the sensitivity studies have been moved to the separate supporting information paper.

Comment

L555: Some caution is warranted when comparing to these field studies, as the VOC were quite different. For example, in the works of Kim and Wolfe, MBO was the dominant VOC.

Response

This is now mentioned in the text.

Changed text:

Old: Kim et al. postulated a missing photolytic HO2 source as the reason for the discrepancy between the measured and modeled HO2 concentration.

New: Kim et al. postulated a missing photolytic HO2 source as the reason for the discrepancy between the measured and modeled HO2 concentration in a 2-methyl-3-buten-2-ol (MBO) dominated environment.

Comment

L600: Z is produced from B-pinene oxidation with an arbitrary yield of 1, so the choice of an equally arbitrary HO2 yield of 6 doesn't seem very unreasonable. There may also be multiple generations of chemistry that are being wrapped up into Z.

Response

We performed an additional simulation to explain the missing HO2 source. Two molecules of HO2 are supposed to be produced from a reactive intermediate together

with a dicarbonyl compound. If the photolytical cleavage of the dicarbonyl compound produces two additional HO2 molecules, the measured HO2* time series can be reproduced by the model. This is now described in detail in chapter 3.5.3.

Comment

Section 4: First paragraph could be shortened a lot. Don't need to re-outline the whole paper, just highlight the key findings. Also in last paragraph, please be more specific about what types of experiments, or what measurements, are needed to pin down this missing HO2 source. SAPHIR is already pretty well-armed.

Response

The section has been shortened and possible additional experiments are now included.

Changed Text

Old:In accordance with the results for ß-pinene presented in this paper we propose a missing photolytic HO2 source as the reason for the underestimation of OH in the model. An additional sensitivity study trying to identify the nature of the HO2 source for the ß-pinene experiment showed that a formaldehyde like photolytic HO2 source is not a reasonable option to explain the measured HO2 and OH levels. Due to the absence of a sufficient amount of photodegradable first generation products at the beginning of the ß-pinene oxidation a photolytic source is not able to produce enough HO2 to explain the measured concentration. A second sensitivity study demonstrated that the addition of RO2 rearrangement reactions releasing HO2 to the model is not a reasonable option either. In case of the RO2 rearrangement reactions the added reaction path competes with the formation of organic nitrates in the model and is thereby causing a strong overestimation of the measured nitrogen oxide concentrations by the model. Additionally a third model run showed that an overestimated yield of organic hydroperoxides can be excluded as the reason for the underestimation of the measured HO2 concentration because the reduction of the HO2 loss is too small. Further studies demonstrated

that an underestimation of the known RO2 interference on the measurements of HO2 can be excluded as the reason for the observed high HO2 concentrations. None of the previously discussed changes in the mechanism as well as the RO2 interference is able to solely explain the deviations between model and measurements. In conclusion, it can be said that the study of the ß-pinene oxidation in SAPHIR as well as several field campaigns showed the lack of understanding of the radical chemistry involved in the OH oxidation of monoterpenes. The identity of the proposed missing HO2 source still remains uncertain. To further elucidate the degradation mechanism for ß-pinene and other monoterpenes more efforts have to be made to quantify degradation products like organic nitrates, hydroperoxides, aldehydes and ketones. Based on this knowledge proceeding investigations determining properties like photolysis rates can be carried out. Potentially the yield of hydroperoxide formation, a sink for HO2 can also have an important influence on the modeled HO2 and OH concentrations.

New: In accordance with the results for ß -pinene presented in this paper we propose an additional HO2 source linked to ß -pinene oxidation products as the reason for the underestimation of OH and HO2 in the model. With additional sensitivity studies it was possible to rule out photolytical processes or rearrangement reactions of RO2 as sole HO2 sources. Also a possible overestimation of the yield of organic hydroperoxides as well an underestimation of the known RO2 interference on the HO2 measurements were excluded as explanations for underestimating HO2 in the model. The gap between measured and modeled HO2* concentration can significantly be reduced modifying the mechanism of Vereecken and Peeters such that the radical intermediate ROO6R2O rearranges rather than being cleaved. The resulting acyl radical produces HO2, CO and a dicarbonyl compound which itself is a photolytical source of HO2 and CO. Still, the exact HO2 formation mechanism remains uncertain. Additional experiments and quantum chemical calculations have to be made to completely unravel the pathway of HO2 formation.

Comment

[Figure]

Technical Comments

L35: delete "to OH"

L180: delete "are"

L221: "interfering"

L436: "formation now depends"

L674: "comprehensively"

Response:

The typos are corrected.

Comment

Figure 1: suggest changing the name of the "MCM mod" run to something like "VP2012." If I read the text right, this is not MCM at all (except for maybe the inorganic chemistry).

Response:

The name has been changed.

Comment

Figure 2: Suggest setting the y scale for the top plot to 0-2.

Response:

The y scaling has been changed.

Comment

Figure 3: Please include a line for the base simulation for comparison.

Response:
* * *
Interactive
comment

The typos are corrected. A base simulation is included in Figure 5. Base simulation charts have been inserted.

Comment

References

Mao, J., Ren, X., Brune, W. H., Van Duin, D. M., Cohen, R. C., Park, J. H., Goldstein, A. H., Paulot, F., Beaver, M. R., Crounse, J. D., Wennberg, P. O., DiGangi, J. P., Henry, S. B., Keutsch, F. N., Park, C., Schade, G. W., Wolfe, G. M., and Thornton, J. A.: Insights into hydroxyl measurements and atmospheric oxidation in a California forest, Atmos. Chem. Phys., 12, 8009-8020, doi: 10.5194/acp-12-8009-2012, 2012.
* * *

---

## Author Response (AR1)

We thank the reviewer for his/her comments. Here are our responses to the specific comments.

Review of "Investigation of the B-pinene photooxidation by OH in the atmosphere simulation chamber SAPHIR", Kaminski et al., ACP (2016)

Summary

This paper analyzes a set of chamber experiments focused on the chemistry of OH + B-pinene. Unlike previous studies, these experiments are done at relatively low VOC and NOx loading, thus better representing true atmospheric conditions. The chamber is also highly instrumented, allowing examination of both radical chemistry and product formation. Interpretation is aided by a box model. The authors find that formation of first-generation products is represented well by a recent theory-based mechanism, but not the MCM. They further identify a major discrepancy between modeled and observed $HO_2$, which they ascribe to an unknown radical source.

The results of these experiments are interesting, novel, and worthy of publication in ACP. The English is verbose/awkward in places, but still understandable. The text is a little long and could be shortened in places. My primary concerns lie in the interpretation of the radical budgets, and in particular the potential for chamber or measurement-based artifacts. Publication is recommended after considering the following.

General Comments

The reviewer is not convinced that a missing $HO_2$ source is the most reasonable hypothesis to fit the observations. Two specific concerns in this regard:

**Comment**

1)      The decay of B-pinene is well described by the base model simulation, and any attempts to artificially increase model HOx degrade this agreement. This is discussed in Section 3.4.3, where it is stated that two independent measurements of both OH and B-pinene (decay) agree well with one another, therefore there seems to be no obvious explanation. One could argue, however, that observations of absolute OH concentration are susceptible to systematic errors, while the decay rate of a VOC is less likely to have such errors. Indeed, the observed decay rate of B-Pinene coupled with the uncertainty in the OH+BPIN rate constant should bracket the range of "reasonable" OH concentrations. The alternative is that an additional process is causing the B-Pinene decay to be artificially slow, but no explanation is given to that end.

**Response**

As explained in chapter 3.4.3 there is a discrepancy between the OH concentration measured directly by LIF and DOAS and the OH concentration as calculated from the measured decay of b-pinene which we cannot explain. However, with any OH dataset the OH production rate corresponds the measured OH destruction rate with in the uncertainty of the instruments. The OH budget is closed. Also, when the measured $HO_2$ concentration in the model is constraint with measured $HO_2$ data, the difference between measured and modelled OH is small. For most of the experiment, the modelled OH concentration is higher than the OH concentration inferred from the VOC measurements and lower than the directly measured data. Therefore, the discrepancy of OH data from direct measurements and OH data inferred from the VOC data is not of importance for the key findings of the paper.

**Comment**

2) HO$_2$* in Fig. 1 climbs steadily throughout the experiment and reaches a maximum around 1300 UTC; indeed, the trend before and after $\beta$-pinene injection is fairly similar. No other species show this behavior. My concern is that this is a chamber wall or instrument artifact, related to either the build-up of oxidation products and/or insolation. It would be nice to see, for example, what the time-progression of J(HCHO) is over this experiment. Does it also show a maximum at ~1300 UTC? The mystery HO$_2$* source reported by Wolfe et al. (2014) showed a marked dependence on solar radiation, so this would provide some support that the two phenomena (whatever their origin) are similar. If J(HCHO) does show a similar trend to HO$_2$*, could the "photolytic source" from molecule "Z" match HO$_2$* if it were set to a constant value instead of being tied to $\beta$-pinene oxidation? Perhaps some long-lived contaminant is injected along with $\beta$-Pinene.

**Response**

The interference of the HO$_2$ data for RO$_2$ radicals has been determined prior to the monoterpene degradation campaign in lab experiments. A sensitivity study (Appendix A) showed that even a doubling the interference would not be enough to explain the measured HO$_2$ values. We agree that the measured HO$_2$* values do not change significantly before and after the VOC injections. A similar time series of HO$_2$* was observed in all $\beta$-pinene experiments, but not in limonene and $\alpha$-pinene experiments which were conducted in the same time period. Therefore this HO$_2$ time series is chemistry related and not caused by artefacts of the instrument. The measured photolysis frequency of formaldehyde J(HCHO) had maximum values between 10:00 AM and 1:30 PM as the measured HO$_2$, but this would be expected from the model, if a photolytic source of HO$_2$ is assumed: In standard experiments HO$_2$* agrees within 15% with the modelled values.

Also we did not observed indications for impurities being injected together with the $\beta$-pinene. The compound was specified with a purity of 99%, and contaminants were detected neither by PTRMS nor by GC/MS.

**Comment**

L10: when referring to "low NOx" in this context, it is better to give NO than NOx mixing ratios since this determines the radical fate.

**Response**

NOx is changed to NO in the text

**Comment**

L18: Factor of 3 low or high?

**Response**

Nopinone concentration is three times higher than measured. The sentence is rephrased.

**Changed text:**

**Old:** The measured OH and HO 2 concentrations were underestimated by up to a factor of two whereas the total OH reactivity was slightly overestimated because of the poor reproduction of the measured nopinone by the model by up to a factor of three

**New**: The measured OH and HO2 concentrations were underestimated by up to a factor of two whereas the total OH reactivity was slightly overestimated because the model predicted a nopinone mixing ratio which was three times higher than measured.

**Comment**

L55: It should also be mentioned that we have since discovered substantial artifacts in some HOx instruments for some conditions (Mao et al., 2012).

**Response**

The study is added to the text.

**Changed text:**

> The newly discovered mechanisms for isoprene and methacrolein, however, can explain only part of the observed high OH concentrations. Another possible reason could be OH interferences in the low pressure laser-induced fluorescence (LIF) instruments that were applied in the above field studies. Artificial OH production was discovered in two similar LIF instruments applying a newly developed chemical modulation technique for OH detection (Mao et al., 2012; Hens et al., 2014; 5 Novelli et al., 2014; Feiner et al., 2016). The interference seems to be related to organic compounds, but the underlying OH formation mechanism is not known. Experimental tests with other type of LIF instruments have not found such interference (Fuchs et al., 2012, 2016; Griffith et al., 2013; Tan et al., 2017), yet it is difficult to draw firm conclusions for past campaigns as long as the reported artefacts (Mao et al., 2012) are not fully understood.

**Comment**

L190: Do these source strengths vary significantly day-to-day? Just curious if this could indicate something about chamber wall aging.

**Response**

The chamber strength of HONO, formaldehyde and acetone can be calculated with good precision from the measured photolysis frequency of NO2 (JNO2), the relative humidity and the temperature using a scaling factor. Prior to each experiment a zero experiment was conducted to determine the scaling factor needed to explain the observed values of these trace compounds. While both the scaling factors for the HONO source and the acetone source did not vary by more than 20% from day to day over the one month period, the scaling factor for formaldehyde increased by a factor of three throughout the campaign.

**Comment**

L246: Suggest ensuring figures and tables are numbered to reflect the order they appear in the text (this may even by ACP policy).

**Response**

Figure numbering is changed.

**Comment**

L261: How does this 70 ppt/h compare to production rate of acetone from BPIN? From figures, it looks like it is relatively small, but it would add confidence to say so.

**Response**

20% to 30% of the total acetone formed in the b-pinene experiment was due to emissions production of the chamber wall.

**Changed text:**

Old: The assumed acetone source strength was typically 70 ppth−1.

New: The assumed acetone source strength was typically 70 ppth−1 which was as large as 20 to 30 % of the total amount of acetone produced in the β-pinene experiments.

**Comment**

L296: More generally, one could just say that the product yield depends on the fate of $RO_2$.

**Response**

This has been changed in the text

**Changed text:**

Old: In principle product yields of nonlinear degradation processes depend on multiple physical and chemical boundary conditions as pressure, temperature, $H_2O$, $O_3$ , VOC, $HO_2$ and NO concentration.

New: In principle product yields of nonlinear degradation processes depend on the fate of $RO_2$ which is governed by multiple physical and chemical boundary conditions such as pressure, temperature, $H_2O$, $O_3$ , VOC, $HO_2$ and NO concentration.

**Comment**

L307: "the yields were normalized to a conversion of 70% of the injected B-pinene." I do not understand what this means; please clarify.

**Response**

The product yields were determined from a linear fit of b-pinene consumed versus the amount of acetone produced. During the course of the experiment the yield of acetone increases due to the increased production from secondary products.  Therefore, only the data of experiments were used for the yield determination when less than 70% of b-pinene was consumed, i.e. when the chemistry of secondary products is still minor relative to the b-pinene chemistry.

**Comment**

L316: Do either the MCM or Vereecken and Peeters mechanisms exhibit an NO dependence in the yield? This could be examined fairly easily with the box model, ramping NO between say 10 ppt and 1 ppb.

**Response**

As proposed, NO was ramped in the model from 10 to 1000 ppt NO. The MCM model suggests an increase from 0.44 at 10 ppt to 0.7 at 1 ppb. The Vereecken model predicts only an increase from 0.21 at 10 ppt NO to 0.26 at 1 ppb.

**Comment**

L418: In these experiments, do measured and modeled $HO_2^*$ agree well?

**Response**

In standard experiments $HO_2^*$ agrees within 15% with the modelled values.

**Comment**

L438: What is meant by "stable?"

**Response**

The sentence has been changed.

**Changed text:**

Old: In contrast to the original MCM 3.2 the primary acetone formation is now depending on two channels, leading to an increase of acetone formation under low NOX conditions, whereas the acetone yield in the MCM 3.2 is fairly stable. More details about the mechanism can be found in Vereecken and Peeters (2012).

New: This leads to an increase of acetone formation at low NO concentrations compared to the MCM 3.2 while the yield of nopinone is predicted to be lower in the model by Vereecken and Peeters (2012).

**Comment**

Section 3.4.4: This section could be shortened significantly by removing details. It seems like a lot of text to present 3 sensitivity studies, none of which explain the missing $HO_2$.

**Response**

This section has been shortened. The sensitivity studies are only mentioned now. The detailed description of the sensitivity studies have been moved to the separate supporting information paper.

**Comment**

L555: Some caution is warranted when comparing to these field studies, as the VOC were quite different. For example, in the works of Kim and Wolfe, MBO was the dominant VOC.

**Response**

This is now mentioned in the text.

**Changed text:**

Old: Kim et al. postulated a missing photolytic $HO_2$ source as the reason for the discrepancy between the measured and modeled $HO_2$ concentration.

New: Kim et al. postulated a missing photolytic $HO_2$ source as the reason for the discrepancy between the measured and modeled $HO_2$ concentration in a 2-methyl-3-buten-2-ol (MBO) dominated environment.

**Comment**

L600: Z is produced from B-pinene oxidation with an arbitrary yield of 1, so the choice of an equally arbitrary $HO_2$ yield of 6 doesn't seem very unreasonable. There may also be multiple generations of chemistry that are being wrapped up into Z.

**Response**

We performed an additional simulation to explain the missing $HO_2$ source. Two molecules of $HO_2$ are supposed to be produced from a reactive intermediate together with a dicarbonyl compound. If the photolytical cleavage of the dicarbonyl compound produces two additional $HO_2$ molecules, the measured $HO_2$* time series can be reproduced by the model. This is now described in detail in chapter 3.5.3.

**Comment**

Section 4: First paragraph could be shortened a lot. Don't need to re-outline the whole paper, just highlight the key findings. Also in last paragraph, please be more specific about what types of experiments, or what measurements, are needed to pin down this missing $HO_2$ source. SAPHIR is already pretty well-armed.

**Response**

The section has been shortened and possible additional experiments are now included.

**Changed Text**

**Old:**

In accordance with the results for β-pinene presented in this paper we propose a missing photolytic $HO_2$ source as the reason for the underestimation of OH in the model. An additional sensitivity study trying to identify the nature of the $HO_2$ source for the β-pinene experiment showed that a formaldehyde like photolytic $HO_2$ source is not a reasonable option to explain the measured $HO_2$ and OH levels. Due to the absence of a sufficient amount of photodegradable first generation products at the beginning of the β-pinene oxidation a photolytic source is not able to produce enough $HO_2$ to explain the measured concentration. A second sensitivity study demonstrated that the addition of $RO_2$ rearrangement reactions releasing $HO_2$ to the model is not a reasonable option either. In case of the $RO_2$ rearrangement reactions the added reaction path competes with the formation of organic

nitrates in the model and is thereby causing a strong overestimation of the measured nitrogen oxide concentrations by the model. Additionally a third model run showed that an overestimated yield of organic hydroperoxides can be excluded as the reason for the underestimation of the measured $HO_2$ concentration because the reduction of the $HO_2$ loss is too small. Further studies demonstrated that an underestimation of the known $RO_2$ interference on the measurements of $HO_2$ can be excluded as the reason for the observed high $HO_2$ concentrations. None of the previously discussed changes in the mechanism as well as the $RO_2$ interference is able to solely explain the deviations between model and measurements.

In conclusion, it can be said that the study of the β-pinene oxidation in SAPHIR as well as several field campaigns showed the lack of understanding of the radical chemistry involved in the OH oxidation of monoterpenes. The identity of the proposed missing $HO_2$ source still remains uncertain. To further elucidate the degradation mechanism for β-pinene and other monoterpenes more efforts have to be made to quantify degradation products like organic nitrates, hydroperoxides, aldehydes and ketones. Based on this knowledge proceeding investigations determining properties like photolysis rates can be carried out. Potentially the yield of hydroperoxide formation, a sink for $HO_2$ can also have an important influence on the modeled $HO_2$ and OH concentrations.

**New:**

In accordance with the results for β -pinene presented in this paper we propose an additional HO2 source linked to β -pinene oxidation products as the reason for the underestimation of OH and HO2 in the model. With additional sensitivity studies it was possible to rule out photolytical processes or rearrangement reactions of RO2 as sole HO2 sources. Also a possible overestimation of the yield of organic hydroperoxides as well an underestimation of the known RO2 interference on the HO2 measurements were excluded as explanations for underestimating HO2 in the model.

The gap between measured and modeled HO2* concentration can significantly be reduced modifying the mechanism of Vereecken and Peeters such that the radical intermediate ROO6R2O rearranges rather than being cleaved. The resulting acyl radical produces HO2, CO and a dicarbonyl compound which itself is a photolytical source of HO2 and CO. Still, the exact HO2 formation mechanism remains uncertain. Additional experiments and quantum chemical calculations have to be made to completely unravel the pathway of HO2 formation.

**Comment**

Technical Comments

L35: delete "to OH"

L180: delete "are"

L221: "interfering"

L436: "formation now depends"

L674: "comprehensively"

**Response:**

The typos are corrected.

**Comment**

Figure 1: suggest changing the name of the "MCM mod" run to something like "VP2012." If I read the text right, this is not MCM at all (except for maybe the inorganic chemistry).

**Response:**

The name has been changed.

**Comment**

Figure 2: Suggest setting the y scale for the top plot to 0-2.

**Response:**

The y scaling has been changed.

**Comment**

Figure 3: Please include a line for the base simulation for comparison.

**Response:**

The typos are corrected. A base simulation is included in Figure 5.  Base simulation charts have been inserted.

**Comment**

References

Mao, J., Ren, X., Brune, W. H., Van Duin, D. M., Cohen, R. C., Park, J. H., Goldstein, A. H., Paulot, F., Beaver, M. R., Crounse, J. D., Wennberg, P. O., DiGangi, J. P., Henry, S. B., Keutsch, F. N., Park, C., Schade, G. W., Wolfe, G. M., and Thornton, J. A.: Insights into hydroxyl measurements and atmospheric oxidation in a California forest, Atmos. Chem. Phys., 12, 8009-8020, doi: 10.5194/acp-12-8009-2012, 2012.

We thank the reviewer for his/her comments. Here are our responses to the specific comments.

Anonymous Referee #2

This paper describes experiments conducted in the SAPHIR chamber in Julich designed to study the photooxidation of beta-pinene. The chamber is well equipped with instrumentation to measure both free radicals (OH, $HO_2$ and $RO_2$) and stable molecules. Consequently, the study focuses mostly on the radical budget, and investigates whether the OH and $HO_2$ rates of production and loss can be determined.

Experimentally, the measured rates of production and loss of OH are found to balance, in contrast to previous studies of isoprene and methacrolein chamber oxidation. However, a model analysis found lower rates of production and destruction, and overall lower radical concentrations. Use of a more detailed model, and measured $HO_2$ concentrations, partially reduced the discrepancy, and sensitivity analyses showed that the measured data could be better represented by introducing an unidentified source of $HO_2$.

Overall, this is a good paper. The experiments and modeling are well described, and attention is paid to uncertainties in the system.

**Comment**

I was a little surprised that only one experiment of the three was analysed in any detail. In fact, no data were shown from two of the experiments (just briefly in the Table). I think this is a serious omission, as these were "normal" experiments, with no added O3, and no extra additions of B-pinene. Would it be possible to include some of these experiments to contrast the results? A wider variation of the NO concentration would be useful. Also, it is unfortunate that no product information is given other than for acetone and nopinone, despite the fact that a PTRMS was used for analysis.

**Response:**

In total we performed three b-pinene experiments in 2012 at NOx concentration lower than 1 ppb. Unfortunately, SAPHIR experiments with the complete set of instruments are quite elaborate, therefore we preferred to repeat experiments at low NO concentrations rather than conduct experiments at elevated NO level. The results of the experiments were similar. In all three experiments, the OH budget was closed, i.e. measured OH production rate was balanced with the measured OH destruction rate. The measured OH and $HO_2$ concentration were higher than predicted from the MCM 3.2. The production of nopinone was underestimated by the MCM3.2 model. The experiment we presented is the only one where there is both OH DOAS and OH LIF data.

The only products we could observe by GC/MS were indeed acetone and nopinone. We were also not able to quantify additional degradation products in PTR-TOFMS measurements, because the concentrations of multiple oxygenated compounds were lower than the detection limit of the PTR-TOFMS and we were lacking the authentic samples for quantification.

**Comment**

Other Comments Line 48. I think it would be better to say that results showed "an incomplete knowledge" rather than "a lack of knowledge". Line 104 (and elsewhere). Insert "such" before "as" i.e. "species such as: : :"

**Response:** This has been changed

Changed text:

The results showed an incomplete knowledge about photochemical oxidation processes under low NO conditions and high BVOC concentrations in these regions (Rohrer et al., 2014)

**Comment**

Line 115 Change "effects" to "affects"

**Response:** This has been changed

**Comment**

Line 167. MCM is a zero-D, or box model (1-D usually refers to a column model with vertical transport).

**Response:** This has been changed

**Comment**

Line 180. Delete "are".

**Response:** This has been changed

**Comment**

Line 221. Inferring should be Interfering?

**Response:** This has been changed

**Comment**

Line 246. Delete one of the double parentheses.

**Response:** This has been changed

**Comment**

Line 253. Change "Caused by" to "As a result of: : :"

**Response:** This has been changed

**Comment**

Line 295. Again, insert "such" before "as".

**Response:** This has been changed

**Comment**

Line 312. Change "then" to "than".

**Response:** This has been changed

**Comment**

Line 346. Clumsy sentence beginning "Caused by: : :".

**Response:** This has been changed

Changed Sentence

Old: Caused by the photochemical reactions of the detected OVOCs plus the unknown species contributing to the background reactivity $RO_2$ and $HO_2$ radicals are produced in SAPHIR, visible in a rise of the $RO_2$* and $HO_2$* concentration.

New: $RO_2$ and $HO_2$ radicals are produced in SAPHIR by photochemical reactions of detected and undetected species visible in a rise of the $RO_2$* and $HO_2$* concentration.

**Comment**

Line 425. "under low NOx conditions". I realize this is somewhat a matter of semantics, and under much discussion at present, but be careful how you classify the NOx environment. With 100 ppt of NO (measured) and up to 20 ppt of $HO_2$, >50% of the $RO_2$radicals will still react with NO. So it is not strictly a low-NOx environment.

**Response:**

Unfortunately, HONO is produced in the illuminated SAPHIR chamber. We reduced the NO mixing ratio by adding ozone to the chamber but we could not reduce the NO concentration to values lower than 100 ppt.

**Comment**

Line 425 or so. I know Vereecken and Peeters ruled this out on the basis of barrier heights, but could a chemically activated BPINAO radical decompose by ring opening, rather than formation of nopinone and CH2OH? It might help to explain the acetone/nopinone dilemma .

**Response**

The decomposition of activated BPINAO followed by ring opening has been proposed by Vereecken and Peeters and is discussed in section 3.4.1. The Vereecken and Peeters model, like the MCM, incorporates the opening of the 4-membered ring, but solves the acetone/nopinone dilemma using ring closure reactions in either the alkylperoxy or the alkoxy radical stage. This is supported by theoretical data, and modelling studies (such as the current work) on $\beta$-pinene oxidation. Barring new evidence, we feel the dilemma is resolved.

**Comment**

Line 427. Remove double parenthesis.

**Response:** This has been changed

**Comment**

Lines 436-438, and 447-450. I think a few more words of clarification might be useful here for people not familiar with the mechanism. As I understand it, the original (as in MCM) fate of BPINCO2 was to react with NO to make acetone predominantly. In the Vereecken and Peeters mechanism, this radical can isomerise under low NOx (to make a bicyclic peroxy radical, which then reacts with NO to make acetone, via a chemically activated alkoxy radical. However, this alkoxy radical can also isomerise (from the aldehyde group) to give different products. A few words describing this train of thought would be helpful. Particularly, be more specific about the radicals involved and how they are reacting.

**Response**

We realize that the description of $\beta$ -pinene is rather short. We have expanded this section to make it clearer. On the other hand, the mechanism proposed by Vereecken et al. is much too complex to be presented in this paper in detail. The main difference between the MCM and the VP2012 mechanism is the ring closure reaction in the early stage of b-pinene oxidation which balances nopinone and acetone formation, whereas the MCM lacks these channels then thus forms either too much nopinone, or too much acetone. The resulting radical ROO6R2O in figure 1 is indeed highly activated. ROO6R2O can either release acetone or isomerize to an acyl radical which can release $HO_2$ and dicarbonyl compound which can be photolyzed to produce another $HO_2$ molecule. If ROO6R2O would completely react via ROO6R8 half of the missing $HO_2$ source could be explained. A discussion on this reaction channel has been included in the section 3.4.4.

**Changed Text**

Old:

An alternative model (Figure 5)) was published by Vereecken and Peeters (2012), including efforts to bring nopinone and acetone model yields in agreement with experimental data.
Based on quantum chemical and theoretical kinetic calculations Vereecken and Peeters proposed a 430 fast ring opening reaction for the intermediate formed by the addition of OH to the double bond of -pinene.

New: l. In the MCM 3.2 mechanism the OH radicals initially add onto the double bonds of β-pinene (Reactions a, b and c in Fig. 1). About 85 % of the molecules are transformed into the tertiary radicals BPINAO1. These radicals add oxygen and form peroxy radicals BPINAO2 (MCM specific designation), which react to nopinone. Acetone is a product of a minor pathway in which the the four-membered ring of β-pinene is broken and BPINCO2 is formed (Reaction b in Fig. 1). An alternative model was published by Vereecken and Peeters (2012). Still, the addition of OH to the external carbon of the double bond forming BPINO1∗ is the main reaction. But in contrast to MCM3.2 Vereecken and Peeters proposed a fast ring opening of BPINAO1∗ based on quantum chemical and theoretical kinetic calculations.

Old: It should be noted that the model by Vereecken and Peeters explicitly marks acetone formation in the current reaction conditions as a valuable metric to calibrate the acetone yield coming from a

specific chemically-activated competition between different reaction channels available to alkoxy radical intermediate ROO6R2O.

New: It should be noted that the acetone formation in the model by Vereecken and Peeters depends on fate of the radical ROO6R2O. This radical can either release acetone or undergo a hydrogen shift to yield radical ROO6R8. Unfortunately,Vereecken and Peeters could not predict the branching of these reactions accurately and were only estimating that acetone cleavage is the dominant reaction. Still, Vereecken and Peeters explicitly mark acetone formation in the current reaction conditions as a valuable metric to verify this branching ratio.

**Comment**

So why should the acetone yield increase at low NOx (line 437), if the MCM predicts only acetone as a product?

**Response**

The dependence of the acetone yield in the Vereecken and Peeters mechanism depends more from the NOx level then the acetone yield in the MCM 3.2 model does. In the Vereecken and Peeters mechanism acetone is from produced from via BPINCO2 two pathways. At low NO concentrations the reaction via ROO6R2O producing acetone is the dominating pathway.

**Comment**

Again, a more detailed analysis of some experiments with varying NO concentrations would have been very useful to diagnose this.

Lines 596-600. As the authors agree, such a large source of $HO_2$ from photolysis of a carbonyl is implausible. But is it? Presumably photolysis leads to the production of 2 radicals. So increasing the photolysis rate by a factor of 3 would work, rather than producing 6 radicals. Is it possible that it is a dicarbonyl similar to glyoxal, which have very fast photolysis rates?

**Response**

We performed an additional simulation to explain the missing $HO_2$ source. Two molecules of $HO_2$ are supposed to be produced from a reactive intermediate together with a dicarbonyl compound. If the photolytical cleavage of the dicarbonyl compound produces two additional $HO_2$ molecules, the measured $HO_2$* time series can be reproduced by the model. This process is now discussed in section 3.4.4.

**Comment**

Line 686. 1-dimensional should be zero-dimensional.

**Response:** This has been changed

**Investigation of the *β*-pinene photooxidation by OH in the atmosphere simulation chamber SAPHIR**

List of relevant changes

M. Kaminski[1,*], H. Fuchs[1], I.-H. Acir[1,**], B. Bohn[1], T. Brauers[1,†], H.-P. Dorn[1], R. Häseler[1], A. Hofzumahaus[1], X. Li[1,***], A. Lutz[2], S. Nehr[1,****], F. Rohrer[1], R. Tillmann[1], L. Vereecken[1], R. Wegener[1], and A. Wahner[1]

Page1 line 20: A passage was added onto the abstract

Page 2 line 20- Page 3 line 30 A passage was added to the introduction to explain artificial OH production in LIF instruments

Page 4 line 20: Instrumentation 2.2.: The passage was changed to clarify radical detection in LIF systems

Page 12 line 14:  Chapter 3.3. Experimental OH Budget analysis: The chapter was rephrased.

Page 13 line 28:  The chapter 3.4.1 was rephrased to describe β-pinene oxidation in detail.

Page 17 line 29: The chapter 3.5.2 'Model sensitivity studies' was shortened. The detailed sensitivity studies were moved to the supporting material.

Page 21 line 1: Chapter 3.5.3 was added to include new model studies which explain the $HO_2$ source.

Page 22 line 25: Within the conclusions the part with the sensitivity studies was shortened, a passage on the new model studies was added.

Page 23: The sensitivity study in the appendix was moved to the supporting material.

Figure 1: The reaction scheme is expanded.

Figure 4: Added to explain the additional $HO_2$ source

Figure 5: New model runs added.

Supporting material is added.

[revised manuscript text omitted]

$$\quad [\mathrm{CH_3COCH_3}]_{corr(i)} = [\mathrm{CH_3COCH_3}]_{corr(i-1)} + \Delta c_{\mathrm{CH_3COCH_3}} + \Delta c_{\underline{RL}\mathrm{RL}} + \Delta c_{\underline{DIL}\mathrm{DIL}} + \Delta c_{S_{\mathrm{CH_3COCH_3}}} \tag{3}$$

$$\Delta c_{\underline{RL}\mathrm{RL}} = [\mathrm{CH_3COCH_3}]_{(i-1)} \cdot [\mathrm{OH}]_{(i-1)} \cdot \Delta t \cdot k_{\mathrm{CH_3COCH_3+OH}} \tag{4}$$

$$\Delta c_{\underline{DIL}\mathrm{DIL}} = [\mathrm{CH_3COCH_3}]_{(i-1)} \cdot \Delta t \cdot k_{\underline{DIL}\mathrm{DIL}} \tag{5}$$

$$\Delta \underline{C} c_{S_{\mathrm{CH_3COCH_3}}} = S_{\mathrm{CH_3COCH_3}} \cdot \Delta t \tag{6}$$

$$S_{\mathrm{CH_3COCH_3}} = a_{\mathrm{CH_3COCH_3}} \cdot J_{\mathrm{NO_2}} \cdot (0.21 + 2.6 \cdot 10^{-2} \cdot RH) \cdot e^{(-2876/T)} \tag{7}$$

$[\mathrm{CH_3COCH_3}]_{\mathrm{corr}}$: corrected acetone concentration

 $\Delta c_{\mathrm{RL}}$: reactive loss

 $\Delta c_{\mathrm{DIL}}$: dilution

[revised manuscript text omitted]

[a] Yield measured by FTIR absorption at 1740 cm$^{-1}$, *and is suspected 7 to include other carbonyl compounds*

**Table 4.** Comparison of measured and modeled product yields from the reaction of $\beta$-pinene with OH radicals for the three $\beta$-pinene injections during the experiment on 27 th August 2012

| Product | Injection | Yield measured | Yield MCM 3.2 | Yield Vereecken and Peeters |
|---------|-----------|----------------|---------------|------------------------------|
| Nopinone | 1 st | 0.28 | 0.53 | 0.27 |
| | 2 nd | 0.37 | 0.61 | 0.28 |
| | 3 rd | 0.35 | 0.65 | 0.30 |
| Acetone | 1 st | 0.20 | 0.07 | 0.37 |
| | 2 nd | 0.24 | 0.16 | 0.47 |
| | 3 rd | 0.36 | 0.21 | 0.49 |